# Priority-Aware Shapley Value

**Kiljae Lee** [1]   **Ziqi Liu** [2]   **Weijing Tang** [2]   **Yuan Zhang** [1]

## Abstract

Shapley values are widely used for model-agnostic data valuation and feature attribution, yet they implicitly assume contributors are interchangeable. This can be problematic when contributors are dependent (e.g., reused/augmented data or causal feature orderings) or when contributions should be adjusted by factors such as trust or risk. We propose **Priority-Aware Shapley Value (PASV)**, which incorporates both hard precedence constraints and soft, contributor-specific priority weights. PASV is applicable to general precedence structures, recovers precedence-only and weight-only Shapley variants as special cases, and is uniquely characterized by natural axioms. We develop an efficient adjacent-swap Metropolis–Hastings sampler for scalable Monte Carlo estimation and analyze limiting regimes induced by extreme priority weights. Experiments on data valuation (MNIST/CIFAR10) and feature attribution (Census Income) demonstrate more structure-faithful allocations and a practical sensitivity analysis via our proposed "priority sweeping".

## 1. Introduction

Recent advances in machine learning have fueled growing interest in *data-centric AI* and *explainable AI*, where a central goal is to quantify how much each training example (data valuation) or each input variable (feature attribution) contributes to a model's performance or predictions (Zha et al., 2025; Mersha et al., 2024). Shapley-type values are a standard tool for *model-agnostic* attribution (Shapley, 1953; Ghorbani & Zou, 2019; Lundberg & Lee, 2017; Kwon & Zou, 2021; Wang & Jia, 2023). A key axiomatization choice behind the Shapley value is that it averages marginal contributions over *all* player permutations, implicitly treating

[1]Department of Statistics, The Ohio State University, OH, USA [2]Department of Statistics and Data Science, Carnegie Mellon University, PA, USA. Correspondence to: Yuan Zhang <yzhanghf@stat.osu.edu>.

*Proceedings of the $43^{rd}$ International Conference on Machine Learning*, Seoul, South Korea. PMLR 306, 2026. Copyright 2026 by the author(s).

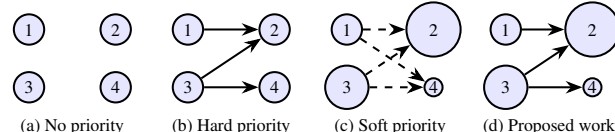

*Figure 1.* Illustration of priority structures. Dashed arrows in (c): existing works on (c) require special precedence structures.

players as interchangeable.

However, in many modern machine learning applications, this exchangeability is questionable for two distinct reasons. First, players can exhibit *precedence relations* in between. For example, training data may be augmented, copied, or otherwise reused, so a downstream contribution can be largely derivative of an upstream source (Aas et al., 2021); likewise, domain or causal knowledge can impose that some features precede others, making certain orders scientifically meaningless (Frye et al., 2021; Janzing et al., 2020). Recent work further highlights that Shapley-type payoff methods can face *copier attack*. This has inspired replication-robust formulations (Han et al., 2022; Falconer et al., 2025) and group-based approaches (Lee et al., 2025) designed to mitigate such manipulations, reinforcing the need to explicitly incorporate precedence into valuation.

Second, practitioners often have *trust/risk considerations*, e.g., some data points may be less trusted for their independence, more costly to curate, or carry higher legal/compliance risk (Wu et al., 2023; Wang & Jia, 2023). Therefore, attribution should reflect two types of *priority*: (i) *hard priority* that rules out orders violating known precedence, and (ii) *soft priority* that adjusts the attribution computation between players under the hard priority infrastructure.

Existing Shapley variants typically address these two aspects in isolation. Precedence-based methods restrict attention to precedence-feasible orders (Faigle & Kern, 1992), while weight-based methods bias the ordering distribution via player-specific weights, but exclusively for special precedence structures (Kalai & Samet, 1987; Nowak & Radzik, 1995). Recent machine learning adaptations apply these ideas to data valuation and feature attribution (Zheng et al., 2025; Frye et al., 2020), but commonly rely on coarse precedence specifications that are easy to handle computationally without offering a unified mechanism to combine *general* precedence constraints with *individual* priorities.

To this end, we propose the **Priority-Aware Shapley Value (PASV)**, a single Shapley-type attribution method that simultaneously captures (i) *hard* precedence constraints among players, and (ii) *soft* player-specific priority weights (Figure 1). PASV computes each player's value by averaging its marginal contribution over precedence-feasible orders, while biasing the order distribution according to the specified priorities. This unified view is designed to be both expressive (supporting general precedence constraints) and practically feasible (supporting scalable computation for large player sets).

Our main contributions are:

- **Unified formulation.** We introduce PASV, a Shapley-type value that simultaneously incorporates general precedence constraints and soft priority weights within one coherent random-order framework.

- **Theory and interpretability.** We provide an axiomatic characterization that singles out PASV as a canonical precedence-respecting, weight-aware rule, and we analyze limiting regimes that clarify how extreme priorities can modify the underlying precedence structure.

- **Scalable computation.** We develop an efficient MCMC-based estimator for sampling precedence-feasible orders under non-uniform weighting, enabling Monte Carlo estimation under general precedence.

- **Practical diagnostic via priority sweeping.** Beyond producing a single allocation, PASV provides an actionable *priority-sweeping* stress test: by varying one player (or group)'s weight while keeping others' weights and the precedence graph fixed, practitioners can quantify the sensitivity to trust/risk considerations, flag contributors whose credit is unstable and may significantly impact the result, and support more robust valuation/attribution decisions.

## 2. Preliminaries

### 2.1. From Shapley Value to Random Order Values

Let $[n] = \{1, \ldots, n\}$ denote the set of players (data points, features). For each subset $S \subseteq [n]$, let $U(S)$ be the reward earned by the joint work of members in $S$: here $U : 2^{[n]} \to \mathbb{R}$ is called *utility function*. The classical royalty sharing problem asks for a principled method to split the total revenue $U([n])$ to each player, considering their contributions on a competitive basis. Shapley (1953) showed that the **Shapley Value (SV)** $\psi_i^{\text{SV}}$ defined as follows is the only payoff method that satisfies four axioms: *efficiency, linearity, null player* and *symmetry* (see Appendix A):

$$\psi_i^{\text{SV}}(U) := \sum_{S \subseteq [n] \setminus \{i\}} \frac{|S|!(n - |S| - 1)!}{n!} \big[U(S \cup \{i\}) - U(S)\big].$$

The Shapley value has a very intuitive alternative version, which this paper builds upon. Imagine that the $n$ players join the game in some order $\pi \in \Pi$, where $\Pi$ is the set of all permutations of $[n]$. Let $\pi^i$ be the set of players that arrived earlier than $i$. Define the **random order value (ROV)** as

$$\psi_i(U) = \mathbb{E}_{\pi \sim p}\big[U(\pi^i \cup \{i\}) - U(\pi^i)\big], \qquad (1)$$

where $p$ is a distribution on $\Pi$. Going forward, we will represent each method we review or propose in terms of their $p$ without repeating (1) (they only differ in their $p$'s).

### 2.2. Hard Priority: Partial Order, Directed Acyclic Graph and Linear Extension

The Shapley value $\psi_i^{\text{SV}}(U)$ is the special case $p = \text{Uniform}(\Pi)$. In presence of priority, we must use a different distribution $p$ that respects the priority relationships.

**Definition 2.1** (Partial order set (poset)). A binary relation $\preceq$ is a **partial order** if it is *reflexive* ($i \preceq i$), *antisymmetric* ($i \preceq j$ and $j \preceq i$ imply $i = j$), and *transitive* ($i \preceq j$ and $j \preceq k$ imply $i \preceq k$). The pair $([n], \preceq)$ is called a **Partially Ordered Set (poset)**. Write $i \prec j$ if $i \preceq j$ and $i \neq j$. [1]

A convenient way to represent $([n], \preceq)$ is via a **Directed Acyclic Graph (DAG)**. Formally, a DAG is a graph on the node set $[n]$ with directed edges, such that for any $(i, j)$ pair, $i \prec j$ if and only if there exists a directed path on the DAG from $i$ to $j$. Throughout the paper, we use the equivalent concepts poset and DAG interchangeably for expediency.

The poset/DAG determines which permutation $\pi$'s are valid.

**Definition 2.2** (Linear extension (LE)). A permutation $\pi \in \Pi$ is a *linear extension* (LE) of the poset $([n], \preceq)$ if $i \prec j$ implies that $i$ appears before $j$ in $\pi$. Let $\Pi^{\preceq}$ denote the set of all linear extensions on $[n]$.

**Precedence Shapley Value (PSV)** (Faigle & Kern, 1992). The PSV replaces the $p \sim \text{Uniform}(\Pi)$ in classical SV by $p \sim \text{Uniform}(\Pi^{\preceq})$, namely,

$$p_{\text{PSV}}(\pi) := \mathbb{1}_{[\pi \in \Pi^{\preceq}]} / |\Pi^{\preceq}|. \qquad (2)$$

### 2.3. Soft Priority: Individual (Soft) Priority Weights

While the PSV is a very natural approach to enforce priority, it equally weights all valid linear extensions. In practice, individual players may vary in their trustworthiness, cost/convenience of data collection, potential legal risk of data use, etc. We are inclined to include trusted/low-risk data earlier in $\pi$. Notice that this consideration cannot be captured by the priority relationship (i.e., the DAG). Therefore, Kalai & Samet (1987) and Nowak & Radzik (1995) introduced individual weights $\lambda := (\lambda_1, \ldots, \lambda_n)$, $\lambda_i > 0, \forall i$

---

[1] It may be cleaner to solely define $\prec$. We define and base our narration on $\preceq$ just to better connect to existing literature.

and *Weighted Shapley Value (WSV)*. WSV is defined for a particular class of DAGs, called *ordered partition*.

**Ordered Partition.** Let $(B_1, \ldots, B_m)$ be a partition of $[n]$, i.e., $B_i \cap B_j = \emptyset, \forall 1 \le i < j \le m$ and $\cup_{i=1}^m B_i = [n]$. The DAG consists of edges $i \prec j$ if and only if $i \in B_r$, $j \in B_{r+1}$ for some $r \in [m-1]$.

In WSV, the permutation $\pi \in \Pi^{\preceq}$ is sampled in a backward fashion. To illustrate, suppose $B_1 = \{1, 2\}, B_2 = \{3, 4, 5\}$. First sample $\pi_5$ (the last element of $\pi$) from candidates $3, 4, 5$ with probabilities $(\lambda_3, \lambda_4, \lambda_5)/(\lambda_3 + \lambda_4 + \lambda_5)$. Suppose 4 is selected for $\pi_5$, then select $\pi_4$ among the remaining 3 and 5 w.p. $(\lambda_3, \lambda_5)/(\lambda_3 + \lambda_5)$, and so on. After finishing $B_2$, select $\pi_2$ between 1 and 2, which also decides $\pi_1$.
*Remark* 2.3. From this example, we see that $\lambda$ acts as a *soft priority* parameter: a larger $\lambda_i$ increases the probability that $i$ appears later in $\pi$. This is meaningful in real applications because attribution is inherently *context dependent*: the marginal gain of adding $i$ can be very different when the coalition already contains other (more trusted, less risky, or more "original") contributors. Thus, increasing $\lambda_i$ **does not (imprudently) directly lower $i$'s payoff; instead it prefers to evaluate $i$ under richer contexts**, which can reduce spurious early credit for noisy/duplicated/high-risk contributors while still allowing genuinely complementary contributors to remain valuable even when delayed in $\pi$.

To formalize this procedure, we define two concepts.
**Definition 2.4** (Feasible set). A subset $S \subseteq [n]$ is *feasible* if $i \in S$ and $j \preceq i$ imply $j \in S$. Denote the collection of feasible sets by $\mathcal{S}^{\preceq} := \{S \subseteq [n] : S \text{ is feasible}\}$.

From Definition 2.4 onward, we slightly abuse notation: while $S$ still denotes a subset of $[n]$, it now conforms to $\preceq$.
**Definition 2.5** (Maximal elements). For $S \in \mathcal{S}^{\preceq}$, the set of *maximal elements*, denoted by $\max_{\preceq}(S)$, is defined as

$$\max_{\preceq}(S) := \{i \in S : \nexists j \in S \text{ s.t. } i \prec j\}.$$

**Weighted Shapley Value (WSV)** (Kalai & Samet, 1987; Nowak & Radzik, 1995). Write $\pi = (\pi_1, \ldots, \pi_n) \in \Pi^{\preceq}$ and $S_t := \{\pi_1, \ldots, \pi_t\}$, define

$$p_{\text{WSV}}(\pi) = \prod_{t=1}^n \frac{\lambda_{\pi_t}}{\sum_{j \in \max_{\preceq}(S_t)} \lambda_j}. \quad (3)$$

While WSV addresses the limitation of PSV, it only applies to special DAGs (ordered partitions), limiting its practicality.

## 3. Our Method

### 3.1. Desired Properties

Section 2 reviewed mainstream existing methods. They either (i) treat all linear extensions uniformly (PSV), capturing

hard priority but no individual weighting; or (ii) incorporate weights under the strong structural assumption of ordered partitions (WSV). Our goal is to design a general approach that combines their strengths while avoiding their limitations. Specifically, we look for a $p^{(\preceq, \lambda)}$ that satisfies the following desiderata:

1. **Hard priority:** $p^{(\preceq, \lambda)}$ is supported on $\Pi^{\preceq}$.

2. **Soft priority:** when $i, j$ are both admissible, their selection probabilities should have ratio $\lambda_i : \lambda_j$.

3. If $\lambda_i \equiv$ constant, $p^{(\preceq, \lambda)}$ reduces to PSV.

4. If DAG is an ordered partition, $p^{(\preceq, \lambda)}$ reduces to WSV.

5. $p^{(\preceq, \lambda)}$ should be computationally feasible.

### 3.2. Priority-Aware Shapley Value (PASV)

Now we propose a $p^{(\preceq, \lambda)}$ that simultaneously captures hard precedence and player-specific weights. Our method is called **Priority-Aware Shapley Value (PASV)**.

$$p^{(\preceq, \lambda)}(\pi) \propto \prod_{t=1}^n \frac{\lambda_{\pi_t} |\max_{\preceq}(S_t)|}{\sum_{k \in \max_{\preceq}(S_t)} \lambda_k}, \quad \text{for } \pi \in \Pi^{\preceq}. \quad (4)$$

From (4), we can see how PASV combines the hard and soft priorities – while restricting $\pi$ to $\Pi^{\preceq}$ enforces the hard priority, the soft priorities of admissible candidates for each $\pi_t$ is adjusted by $\lambda_i$'s, thus reflecting the soft priority. When $\lambda_i$'s are all equal, (4) becomes $p^{(\preceq, \lambda)}(\pi) \propto 1$, and PASV recovers PSV. When the DAG is an ordered partition, it can be shown that $p^{(\preceq, \lambda)}(\pi) = \prod_{t=1}^n \frac{\lambda_{\pi_t}}{\sum_{k \in \max_{\preceq}(S_t)} \lambda_k}$, thus recovering WSV. Formally, we have the following results.

**Proposition 3.1** (Reduction to PSV). *If $\lambda_1 = \cdots = \lambda_n$, then $p^{(\preceq, \lambda)}(\pi) = p_{\text{PSV}}(\pi), \forall \pi \in \Pi^{\preceq}$.*

**Proposition 3.2** (Reduction to WSV). *If the DAG is an ordered partition, then $p^{(\preceq, \lambda)}(\pi) = p_{\text{WSV}}(\pi), \forall \pi \in \Pi^{\preceq}$.*

*Remark* 3.3. When developing a general $p^{(\preceq, \lambda)}$, one may be tempted to directly apply the formulation of WSV (3) to general DAGs, but the resulting distribution does not reduce to PSV for equal weights. The $|\max_{\preceq}(S_t)|$ factor on the numerator of (4) is crucial for ensuring Proposition 3.1.

PASV can also be efficiently computed. We relegate the algorithm details to the dedicated Section 4.

### 3.3. Axiomatization of PASV

Like SV, PSV and WSV, our PASV can also be naturally derived from a set of axioms. Here, we provide readers an outline as a preview of this axiomatization. Notice that it contains acronyms and jargon that will be defined later.

- Axioms E + L + NP + M + MS $\Rightarrow$ PROV (a big family)

- Focus on "SCF form" to narrow down PROV

- Add axioms WP + EWU to align with boundary cases

Finally, PROV + SCF + WP + EWU $\Rightarrow$ PASV. Next, we carry out and elaborate this outline.

### 3.3.1. PRECEDENCE RANDOM ORDER VALUE (PROV)

Recall that the classical Shapley value can be rewritten as a random order value (ROV). In presence of priority, we should of course constrain our consideration to $p$ distributions supported not on all of $\Pi$, but on the subset $\Pi^{\preceq} \subseteq \Pi$ that respects priority.

**Definition 3.4** (Precedence random order value (PROV)). A value $\psi$ is called a **precedence random order value (PROV)** if there exists some $p$ supported on $\Pi^{\preceq}$, s.t. $\psi_i(U) = \mathbb{E}_{\pi \sim p}\big[U(\pi^i \cup \{i\}) - U(\pi^i)\big]$ for all $U$ and $i$.

While it may be natural to treat PROV as a basic assumption or an axiom, we will show that in fact PROV can be derived from five more basic axioms. These axioms include three Shapley axioms: Efficiency (E), Linearity (L) and Null player (NP) (see Appendix A); plus two more axioms: Monotonicity (M) and Maximal support (MS).

**Axiom 3.5** (Monotonicity). If $U$ is *monotone*, i.e., $S \subseteq T \Rightarrow U(S) \le U(T)$ for all $S, T \subseteq [n]$, then every player receives a nonnegative payoff: $\psi_i(U) \ge 0$ for all $i \in [n]$.

Weber (1988) showed that axioms E + L + NP + M $\Rightarrow$ ROV. Since PROV is a special ROV, it naturally inherits these axioms. To present axiom MS, we first define *elementary game* – it has long been used for analyzing Shapley's values.

**Definition 3.6** (Elementary game). For a given subset $T \subseteq [n]$, the *elementary game* $u_T \in \Gamma$ on $T$ is defined by $u_T(S) := \mathbb{1}\{T \subseteq S\}$ for $S \subseteq [n]$.

We focus on elementary games for feasible $T$, i.e., $T \in \mathcal{S}^{\preceq}$ (recall Definition 2.4). The precedence relation singles out the *maximal* elements of $T$ as sole the recipients of credit. This motivates the following axiom.

**Axiom 3.7** (Maximal support). For any elementary game $u_T$ with $T \in \mathcal{S}^{\preceq}$, if player $i$ is not maximal in $T$, i.e., $i \notin \max_{\preceq}(T)$, then $\psi_i(u_T) = 0$.

**Theorem 3.8** (Axiomatization of PROV). *A value $\psi$ is PROV $\Leftrightarrow$ it satisfies axioms E + L + NP + M + MS.*

### 3.3.2. STATE-CHOICE FACTORIZATION (SCF)

The PROV as defined in Definition 3.4 is a huge family, and using the most general PROV directly is computationally challenging. We therefore restrict attention to a broad and structured family of precedence-respecting order distributions that factorize along the chain of feasible sets induced by a linear extension. Conceptually, this is the precedence-constrained analogue of stage-wise models for rankings and permutations, such as the Plackett–Luce (Luce et al., 1959; Plackett, 1975) and multistage ranking families (Fligner & Verducci, 1988). To this end, we consider the family of distributions on $\Pi^{\preceq}$ of the form

$$p(\pi) \propto \prod_{t=1}^{n} w_{\preceq,\lambda}(\pi_t; S_t), \quad \text{for } \pi \in \Pi^{\preceq} \qquad (5)$$

for some $w_{\preceq,\lambda}$. This form (5) is the precedence-constrained analogue of Plackett-Luce: the likelihood of some $\pi$ depends on $\pi$ only through the sequence of feasible candidates $\{S_t\}_{t=1}^{n}$, and the chosen candidate at this time. It is often convenient to decompose $w$ as a *scale* ("state"): $s_{\preceq}(S_t) := \sum_{k \in \max_{\preceq}(S_t)} w_{\preceq,\lambda}(k; S_t)$, where we shall further assume that $s_{\preceq}(S_t)$ does not depend on $\lambda$, which is true for WSV; and a *normalized priority weighting* ("choice"): $c_{\preceq,\lambda}(i; S_t) := w_{\preceq,\lambda}(i; S_t)/s_{\preceq}(S_t)$.

**Definition 3.9** (State–Choice Factorization (SCF)). For a poset $([n], \preceq)$, define the prefix (remaining) set $S_t = \{\pi_1, \ldots, \pi_t\}$ as a chain of feasible sets ($t = n, n-1, \ldots, 1$). A **state-choice factorization (SCF)** assigns an unnormalized weight to $\pi$ by multiplying:

(i) a **state factor** $s_{\preceq}(S_t)$ that captures global bias induced by the precedence structure;

(ii) a **choice factor** $c_{\preceq,\lambda}(\,\cdot\,; S)$ that encodes soft priority weighting of admissible candidates $\max_{\preceq}(S)$; it is the entry point for the individual weights $\lambda$.

Formally, an SCF $p$ has the following form

$$p(\pi) \propto \prod_{t=1}^{n} s_{\preceq}(S_t)\, c_{\preceq,\lambda}(\pi_t; S_t), \quad \text{for } \pi \in \Pi^{\preceq}, \quad (6)$$

where $\sum_{i \in \max_{\preceq}(S_t)} c_{\preceq,\lambda}(i; S_t) = 1$.

The merits of SCF are three-fold:

- It remains flexible enough to capture a broad range of precedence-respecting random-order models, containing PSV and WSV as special cases.

- It preserves a clear interpretation of $\lambda$ weights as a soft priority parameter;

- Its factorization form enables efficient MCMC sampling procedures for generating $\pi$.

### 3.3.3. BOUNDARY CASE AXIOMS

We desire our method to contain PSV and WSV as special cases. This is characterized by the following two axioms.

**Axiom 3.10** (Weight Proportionality). For every feasible set $S \in \mathcal{S}^{\preceq}$ and every $i, j \in \max_{\preceq}(S)$, $\frac{c_{\preceq,\lambda}(i;S)}{c_{\preceq,\lambda}(j;S)} = \frac{\lambda_i}{\lambda_j}$.

Axiom WP requires that at each step the probability of choosing each admissible candidate $i$ is proportional to $\lambda_i$.

**Axiom 3.11** (Equal-Weight Uniformity). If $\lambda_i \equiv c$ for all $i \in [n]$, then the induced distribution is uniform on $\Pi^{\preceq}$.

Axiom 3.10 recovers WSV and Axiom 3.11 recovers PSV.

### 3.3.4. AXIOMATIZATION OF PASV

**Theorem 3.12.** *Suppose $p$ is a PROV as in Definition 3.4 satisfying the SCF form as in (6). Then the only $p$ that also satisfies Axioms 3.10 and 3.11 is PASV as defined in (4).*

### 3.4. Limiting Cases of the Priority-Aware Distribution

This section studies what happens to the PASV order distribution when a player weight becomes extreme. The goal is to clarify how the *soft* priorities encoded by $\lambda$ interact with the *hard* precedence constraints $\preceq$ (equivalently, the DAG). In PASV (4), $\lambda$ affects the distribution only through the choice factor $c_{\preceq,\lambda}(i;S)$ over the currently admissible (maximal) set $\max_{\preceq}(S)$. For $i \in \max_{\preceq}(S)$, we have

$$c_{\preceq,\lambda}(i;S) = \frac{\lambda_i}{\lambda_i + \sum_{k \in \max_{\preceq}(S) \setminus \{i\}} \lambda_k}. \quad (7)$$

If $\max_{\preceq}(S) = \{i\}$, changing $\lambda_i$ has no effect. When $|\max_{\preceq}(S)| \geq 2$ all other weights are held fixed, (7) implies $c_{\preceq,\lambda}(i;S) \to 1$ as $\lambda_i \to \infty$ and $c_{\preceq,\lambda}(i;S) \to 0$ as $\lambda_i \to 0^+$. In other words, **extreme weights induce an (almost) deterministic tie-break among simultaneously admissible players**: $\lambda_i \to \infty$ selects $i$ whenever it is maximal, i.e., pushing $i$ as *late* as allowed in $\pi$ (in the forward order), while $\lambda_i \to 0^+$ pulls $i$ as *early* as possible in $\pi$.

Interestingly, in some cases, extreme weights can simply translate into new DAG edges. We showcase two scenarios: (i) the extreme-weight player is globally maximal, and (ii) DAG is an ordered partition. Figure 2 illustrates two examples. Importantly, we emphasize the following takeaways.

- These new edges will **not** create cycles; and

- Setting weights in whatever way will **not** delete, rewire or reverse existing edges in the original DAG.

Formally, we have the following theoretical results.

**Proposition 3.13.** *Fix a poset $([n], \preceq)$ and $i \in \max_{\preceq}([n])$, let $\preceq'$ be the partial order obtained by adding $j \prec i$ for all*

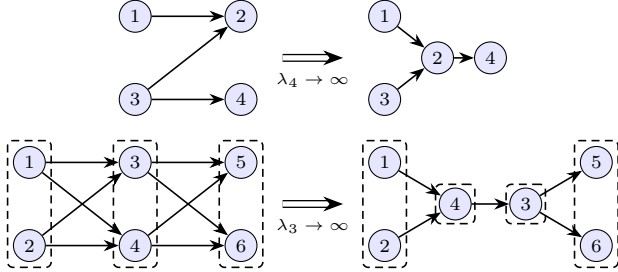

*Figure 2.* Extreme $\lambda$ may translate into DAG edges: Top: sending $\lambda_4 \to \infty$ adds an edge $2 \to 4$ (on the right we omitted the edge $3 \to 4$ as it is implied by the path $3 \to 2 \to 4$). Bottom: sending $\lambda_3 \to \infty$ adds an edge $4 \to 3$, forming a refined layer structure.

$j \in \max_{\preceq}([n]) \setminus \{i\}$. *Then for any $\lambda'$ satisfying $\lambda'_j = \lambda_j$ for all $j \neq i$ (with arbitrary $\lambda'_i > 0$), and for every $\pi \in \Pi$,*

$$\lim_{\lambda_i \to \infty} p^{(\preceq,\lambda)}(\pi) = p^{(\preceq',\lambda')}(\pi) \cdot \mathbb{1}_{[\pi \in \Pi^{\preceq'}]}.$$

**Proposition 3.14.** *Assume $\preceq$ is an ordered partition $(B_1, \ldots, B_m)$. Fix a layer $B \in \{B_1, \ldots, B_m\}$ and a nonempty subset $G \subseteq B$. Fix weights $(\lambda_j)_{j \notin G}$ and consider the regime in which $\lambda_i = \bar{\lambda}$ for all $i \in G$ while $\bar{\lambda} \to \infty$. Let $\preceq'$ represent the refined ordered partition obtained by splitting $B$ into two ordered, consecutive layers $(B \setminus G, G)$. Then for any $\lambda'$ satisfying $\lambda'_j = \lambda_j$ for all $j \notin G$ and $\lambda_i = \widetilde{\lambda}$ for $i \in G$ with arbitrary $\widetilde{\lambda} > 0$, and for every $\pi \in \Pi$,*

$$\lim_{\bar{\lambda} \to \infty} p^{(\preceq,\lambda)}(\pi) = p^{(\preceq',\lambda')}(\pi) \mathbb{1}_{[\pi \in \Pi^{\preceq'}]}.$$

More broadly, however, extreme weights are *not* equivalent to adding a fixed set of edges in general. The key reason is that the tie-break is triggered only when the extreme-weight player becomes maximal, and *whether/when* this happens depends on the realized history of the (backward) construction of $\pi$. Figure 3 gives a counterexample.

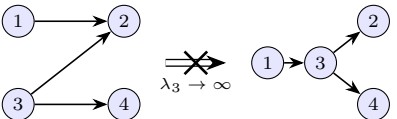

*Figure 3.* Extreme $\lambda$ may not always translate into a new DAG edge: if $\pi_4 = 2$, then 1 and 4 are both eligible for $\pi_3$; the (only possible) edge to add $1 \to 3$ would incorrectly deny 1's candidacy.

**Priority sweeping.** Beyond producing a single valuation, PASV enables a handy sensitivity/robustness test w.r.t. $\lambda$. Fixing the DAG and a baseline $\lambda$, for each player $i$, our **priority sweeping** tracks how the valuation of $i$ changes as $\lambda_i$ varies in $(0, \infty)$. This would tell us which players' values are stable versus sensitive to its weight.

## 4. Computation Algorithm for PASV

Recall that PASV is defined in an ROV form as in (1), where $p = p_{\text{PASV}}$ as in (4). While the expectation in (1) can be

easily approximated by a Monte Carlo sampling with i.i.d. $\pi \sim p_{\text{PASV}}$, the main computational challenge lies exactly in how to sample $\pi$ from the distribution $p_{\text{PASV}}$, because the number of valid $\pi$'s grows exponentially with $n$. Practical computation therefore typically relies on Markov Chain Monte Carlo (MCMC) strategies.

**Adjacent-swap M-H on $\Pi^{\preceq}$.**  The common approach to MCMC on $\Pi^{\preceq}$ is via *adjacent swaps* of incomparable neighbors (Karzanov & Khachiyan, 1991; Bubley & Dyer, 1999). We adopt this method but generalize the uniform distribution to $p^{(\preceq,\lambda)}(\cdot)$. To start, it is not difficult to find one valid $\pi \in \Pi^{\preceq}$ as the initialization (Kahn, 1962). Then in each iteration, sample $k \in [n-1]$ at random. If $(\pi_k, \pi_{k+1})$ are *incomparable*, i.e., neither of $\pi_k, \pi_{k+1}$ precedes the other, propose $\pi'$ by swapping them; otherwise, stay at $\pi$. The Metropolis–Hastings (M-H) acceptance probability is

$$\alpha := \mathbb{P}(\pi \to \pi') = \min\left\{1, p^{(\preceq,\lambda)}(\pi') / p^{(\preceq,\lambda)}(\pi)\right\}. \quad (8)$$

Although $p^{(\preceq,\lambda)}(\cdot)$ is defined as a product over $n$ steps, the acceptance ratio in (8) can be evaluated using only local information around chosen position $k$.

**Lemma 4.1.** *Fix a poset $([n], \preceq)$ and weights $\lambda$. Let $\pi \in \Pi^{\preceq}$ and fix $k \in [n-1]$ such that $\pi_k$ and $\pi_{k+1}$ are incomparable. Let $\pi'$ be the permutation obtained by swapping $(\pi_k, \pi_{k+1})$ in $\pi$. Then we have*

$$\frac{p^{(\preceq,\lambda)}(\pi')}{p^{(\preceq,\lambda)}(\pi)} = \frac{\left(\sum_{i \in M_k} \lambda_i\right)/|M_k|}{\left(\sum_{j \in M'_k} \lambda_j\right)/|M'_k|}, \quad (9)$$

*where $M_k = \max_{\preceq}(S_k)$, $M'_k = \max_{\preceq}(S'_k)$, $S_k = \{\pi_t\}_{t \in [k]}$, and $S'_k = \{\pi_t\}_{t \in [k-1] \cup \{k+1\}}$.*

Algorithm 1 describes our algorithm for sampling $\pi \sim p_{\text{PASV}}$. Based on Algorithm 1, PASV can be efficiently approximated by a Monte Carlo on (1) without difficulty.

# 5. Experiments

## 5.1. PASV for Data Valuation

We evaluate PASV in a data market setting with explicit *data lineage* (some providers' training examples are derived or copied from others), comparing it against existing methods.

**Setup: data and utility.**  We use MNIST (LeCun et al., 2002) and CIFAR10 (Krizhevsky et al., 2009), both containing 10 classes, with a $k$-NN classifier (MNIST: pixel features; CIFAR10: pretrained ResNet-18 embeddings (He et al., 2016)). We fix a balanced test set of size 1000 (100 per class) and define the utility of a training subset $S$ as test accuracy, $U(S)$. We consider eight data providers, each contributing 100 training data points ($n = 800$ players (data points) in total):

---

**Algorithm 1** M-H algorithm for sampling $\pi \sim p^{(\preceq,\lambda)}(\cdot)$.

**Input:** poset $([n], \preceq)$; weights $\lambda$; number of samples $N_{\text{MC}}$; burn-in $B$; thinning $\tau$; index sampler $f$ on $[n-1]$.
**Initialize** $\pi^{(0)} \in \Pi^{\preceq}$; Set of sampled $\pi$'s: $\mathcal{T} \leftarrow \emptyset$
**for** $t = 1$ **to** $B + \tau(N_{\text{MC}} - 1) + 1$ **do**
  Draw $k \sim f$; set $\pi \leftarrow \pi^{(t-1)}$
  **if** $\pi_k$ and $\pi_{k+1}$ are incomparable **then**
    $\pi' \leftarrow (\pi$ with $(\pi_k, \pi_{k+1})$ swapped$)$
    Compute $(S_k, S'_k, M_k, M'_k)$ and $\alpha$, c.f. (8), (9)
    With prob. $\alpha$, set $\pi^{(t)} \leftarrow \pi'$; otherwise set $\pi^{(t)} \leftarrow \pi$
  **else**
    $\pi^{(t)} \leftarrow \pi$
  **end if**
  **if** $t > B$ **and** $(t - B - 1) \bmod \tau = 0$ **then**
    Append $\pi^{(t)}$ to $\mathcal{T}$
  **end if**
**end for**
**return** $\mathcal{T}$

---

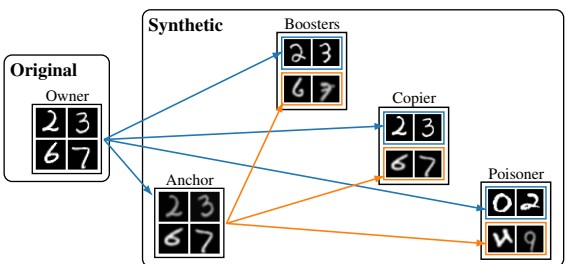

*Figure 4.* The DAG used for MNIST/CIFAR10. Only one Booster is shown. Here, (for example) a blue edge from Owner to half of Copier represents $100 \times 50$ edges between their members.

- **Owner:** original labeled data points.

- **Anchor:** classical augmentation applied to Owner (e.g., rotation/blur); producing synthetic data points that downstream providers may reuse.

- **Four Boosters:** four generative model families, GAN (Goodfellow et al., 2020), DDPM (Ho et al., 2020), DDIM (Song et al., 2020), and FM (Lipman et al., 2022) that produce synthetic data points. Each Booster contributes 50 samples from a genAI model trained on Owner and another 50 samples trained on Anchor.

- **Copier:** copies half from Owner and half from Anchor.

- **Poisoner:** copies the same way as Copier but randomly flips labels (harmful reuse).

**Data lineage and DAG.**  We encode data lineage as a precedence DAG over *blocks*. Owner and Anchor each form a block. For every reuse-based provider (4 Boosters, Copier, Poisoner), we split its data into two blocks: Owner-derived vs. Anchor-derived. This yields a 14 block-DAG (which is not an ordered partition), illustrated in Figure 4.

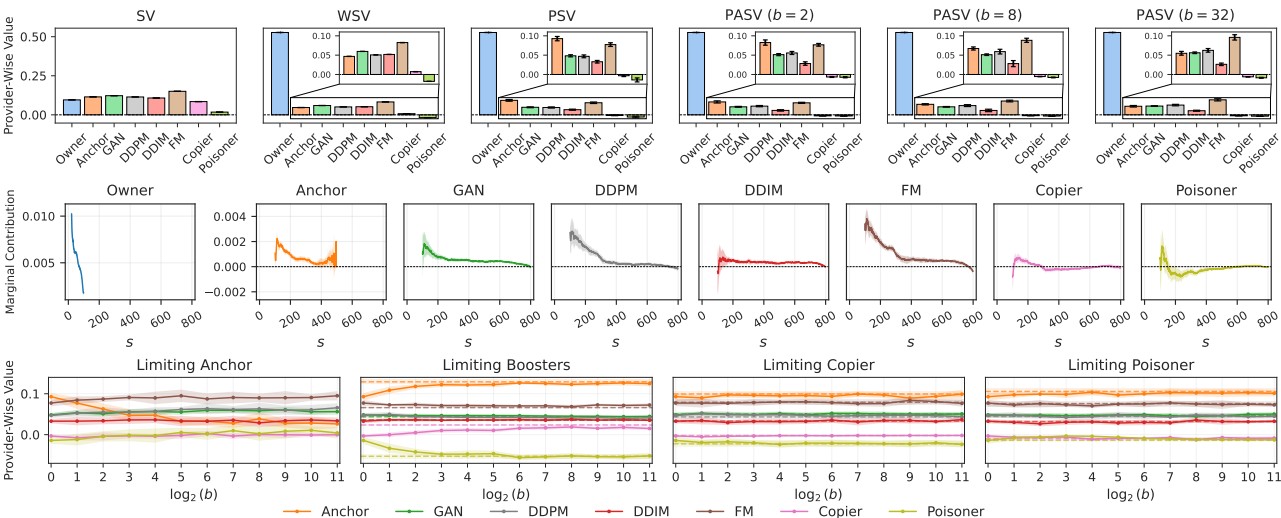

*Figure 5.* MNIST results. **Top:** Provider-level values (summed value for each provider (Lee et al., 2025); 10 repetitions, report mean & $\pm 1$ std. bars; **Middle:** Marginal contributions: $\mathbb{E}[U(\pi^i \cup \{i\}) - U(\pi^i)|\|\pi^i\| = s]$ vs $s$, c.f. (1), $p := \text{Uniform}(\Pi^{\prec})$; **Bottom:** Priority sweeping: $c \in \{(0,1,0,0,0), \ldots, (0,0,0,0,1)\}, b \in \{2^k : k \in [11]\}$, Owner not swept (alone in maximal set, weight has no effect).

**Benchmarks.** (i) **classical SV** (Ghorbani & Zou, 2019); (ii) **WSV:** approximate the DAG by a two-layer ordered-partition: Original (Owner) $\prec$ Synthetic (Anchor, Boosters, etc) (Zheng et al., 2025); (iii) **PSV** (Faigle & Kern, 1992).

**Soft priority weights.** We set individual weights according to their provider type as $\lambda = b^c$, with a shared base $b > 1$ and exponents $c = (c_{\text{Owner}}, c_{\text{Anchor}}, c_{\text{Booster}}, c_{\text{Copier}}, c_{\text{Poisoner}})$.

**Main comparison.** We first conduct a *main comparison*, in which $c = (0, 1, 0, 2, 2)$ and $b \in \{2, 8, 32\}$, mildly penalizing Anchor and more strongly penalizing Copier/Poisoner. Here, we use weights to softly incorporate our prior knowledge of *usefulness*, e.g., anticipating that the test set might not necessarily feature rotated images as seen in Anchor, our $\lambda$ favored Boosters over Anchor, since Boosters contain data generated directly based on Owner without excess rotation.

Row 1 of Figure 5 reports *provider-level* results for MNIST (CIFAR10 in Appendix E.3). SV over-credits data reuse (e.g., Copier), as it ignores precedence. WSV favors Owner by forcing "Original $\prec$ Synthetic" but still mixes up among synthetic providers due to its structural limitation (Anchor is less credited, positive values for Copier). PSV better preserves credit for upstream sources and can reveal negative value for Copier. PASV then provides a more flexible controlled adjustment: increasing $b$ progressively suppresses the penalized types (Anchor/Copier/Poisoner) while respecting the same reuse constraints as PSV.

To better understand the effect of priority sweeping in PASV relative to PSV, we examine how a data point's marginal gain depends on its entry position in $\pi$. In PASV, data points with larger $\lambda_j$ are more likely to be evaluated at later posi-

tions, when permitted by the DAG (see Remark 2.3). We therefore plot the average marginal gain conditional on entering at each position $s$ for each data provider. Row 2 of Figure 5 shows the marginal-gain curves and offers insights (c.f. caption of Figure 5). Anchor and 4 Boosters show similar trends: the earlier in $\pi$ the more significant the recognition of their positive contributions, and FM showed the most prominent trend. These position-dependent patterns explain the difference between PSV and PASV observed in Row 1. PASV tends to evaluate data from Anchor, Copier, and Poisoner in richer coalition contexts and thus places more weight on their tail-regime marginal gains, leading to more diminished final valuations.

Interestingly, Copier/Poisoner both show a "positive spike" for small $|\pi^i|$; this is because when $|\pi^i| \approx 100$, the sampled size from each class (on average just 10) may be unstable; in this situation, adding copied or even noisy data may still benefit the numerical stability of the classifier: when a class has few members present in a small $\pi^i$, adding Copier data can help $k$-NN avoid drawing neighbors from other classes.

**Sensitivity analysis.** Row 3 of Figure 5 shows how provider-level valuations vary as $b$ grows for each of Anchor/Booster/Copier/Poisoner with their own indicator $c$, c.f. caption of Figure 5. Applying our theoretical result in Section 3.4, we predict that as $b \to \infty$, for each of Boosters/Copier/Poisoner, we can compute a PSV with the DAG with added edges incoming from other providers (e.g., the analysis for Boosters sets $c = (0, 0, 1, 0, 0)$, then $b \to \infty$ is equivalent to adding edges $\{\text{Copier}, \text{Poisoner}\} \to \text{Boosters}$). In Row 3 plots, we used dashed lines to mark the PASV computed under the new DAG with added edges, and they well-corroborate observed PASV values for large $b$, vali-

dating our theory's prediction. However, this simplified translation does not apply to Anchor for the same reason as the example in Figure 3.

### 5.2. Application to Feature Attribution

While priority relations are usually considered for data Shapley, in feature Shapley, domain knowledge may require certain variables to be considered before others (e.g., causal ancestors before descendants) (Frye et al., 2020). Here we test PASV in an attribution setting with precedence constraints encoded by a general DAG and focus on the attribution sensitivity to: (i) DAG; and (ii) priority weights.

**Setup.** We follow Frye et al. (2020) on the UCI Census Income (Adult) dataset (Asuncion et al., 2007), predicting whether annual income exceeds \$50K. We remove `fnlwgt` and `education-num` to avoid redundancy, leaving $n = 12$ features as players. We train a two-hidden-layer MLP with 100 units per layer.

**Utility function.** For a predictor $f$, let $f_y(x)$ denote the predicted probability of assigning an input $x$ to the (true) class $y$. For any subset of features $S \subseteq [n]$, we first use a $k$-NN imputation procedure (detailed steps in Appendix F.1) to estimate $u_S(x, y) = \mathbb{E}_{x' \sim p(x'|x_S)}[f_y(x_S \oplus x'_{S^c})]$ (Janzing et al., 2020; Sundararajan & Najmi, 2020; Frye et al., 2021). Then we can define the utility function as $U(S) = \mathbb{E}_{(x,y) \sim p(x,y)}[u_S(x, y)]$ and approximate it by empirically sampling $(x, y)$ from the original data.

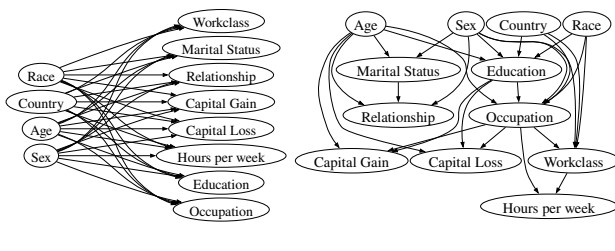

*(a)* Ordered Partition      *(b)* General DAG

Figure 6. Two DAG specifications over the 12 income features.

**DAG specification.** Figure 6 compares two DAGs encoding which feature orders are admissible. The first is a coarse ordered partition used by Frye et al. (2020), where four demographic variables precede the rest (Figure 6a). The second is a more detailed precedence DAG over the same 12 features (Figure 6b) that adds AI-suggested additional relations based on common sense and domain knowledge (e.g., education $\prec$ occupation); details in Appendix F.3.

**Experimental procedure.** For each DAG, we first compute PASV with uniform weights $\lambda_i \equiv 1$, which reduces to PSV. For the ordered-partition DAG, this also coincides with

the uniform-weight WSV; for the general DAG, computing PSV/PASV requires the MCMC sampler in Section 4. Next, we perform a *priority sweeping*: for each feature $i$, vary $\lambda_i \in \{2^k : k \in [-8 : 8]\}$ while holding $\lambda_j \equiv 1, \forall j : j \neq i$.

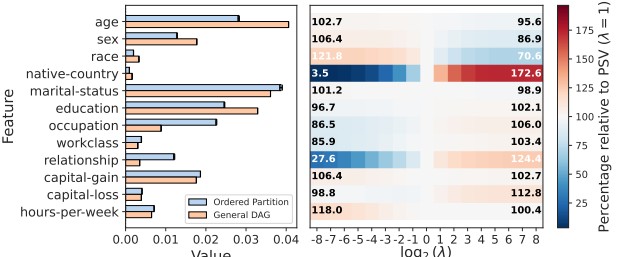

Figure 7. Results of feature attribution for Census Income dataset. (**Left**) PSV (=PASV) under the two DAGs with uniform weights. (**Right**) Sensitivity analysis using priority sweeping.

**Results.** Figure 7 highlights two takeaways. First, the left panel shows that **DAG matters a lot:** with the same utility function, the PSV valuation changes noticeably between ordered partition and the richer DAG, indicating that using a coarse DAG (e.g., ordered partition) may significantly alter results. Second, the priority sweeping with our PASV in the right panel **provides a crucial sensitivity diagnosis**. For example, several demographic variables (`age`, `sex`, `race`) decrease when delayed (large $\lambda$), whereas others (e.g., `native-country`, `relationship`, `occupation`) increase, and some features exhibit non-monotone responses. This is consistent with the intuitive understanding in Section 5.1: moving a feature later changes the set of variables already present when its marginal contribution is evaluated, so the effect depends on whether the feature is redundant with, or complementary to, its typical competitors under the assumed DAG. Overall, PASV's priority sweeping provides a practical diagnostic tool for identifying which attributions are robust to priority weighting (e.g., `martial-status`, `capital-loss`, etc) and which are unstable (most noticeably `native-country` and `relationship`).

## 6. Computational Cost and DAG Sensitivity

We now address two practical deployment questions: how expensive is PASV on larger player sets, and how sensitive is the result to the DAG?

### 6.1. Computational Cost

**Overview.** PASV's computation has two distinct stages: 1. a burn-in stage that runs the adjacent-swap M-H chain until stationarity, no utility evaluations; 2. a Monte Carlo stage that retains thinned samples of player permutations, evaluates $U(S)$ along the resulting prefix sets, and use the induced marginal contributions to approximate PASV. In

the data-valuation experiment of Section 5.1, the utility involves training and evaluating a downstream classifier, and the Monte Carlo stage dominates. Under inexpensive utilities, however, the burn-in stage can be non-negligible.

**Stage 1: mixing time.** We assess mixing via a pairwise order discrepancy $D_t = \max_{i,j \in [n]} |\mathbb{P}^t_{\pi_0}(i \prec_\pi j) - \mathbb{P}^\star(i \prec_\pi j)|$, where $i \prec_\pi j$ means $i$ appears before $j$ in $\pi$, $\mathbb{P}^\star$ is the stationary distribution, and the *empirical mixing time* is $\min\{t : D_t < 1/4\}$. We compute $\mathbb{P}^\star$ exactly by adapting the backward-removal recursion of Kangas et al. (2016) to replace unweighted extension counts with a $\lambda$-weighted mass; see Appendix D.1.1. Across four synthetic poset families in Talvitie et al. (2017) with $\lambda$ ranging from a uniform baseline to $\lambda_i \sim \text{Unif}(1, R)$ with $R$ up to 1024, Figure 8 in Appendix D.1.1 shows that the empirical mixing time tracks $O(n^2)$ (Talvitie et al., 2017) rather than the worst-case bound, and that non-uniformity of $\lambda$ has limited effect on mixing.

**Stage 2: Monte Carlo accuracy of direct MC estimator.** We benchmark on a closed-form sum-of-unanimity utility whose true PASV admits a tractable expression (Appendix D.1.2). On this benchmark, the direct Monte Carlo estimator attains a small relative error using only a few hundred retained permutations, and remains tractable up to $n = 8192$ players under moderate compute (Figure 9).

**An (optional) Regression MSR proxy for stage 2.** The direct Monte Carlo estimator is simple and broadly applicable, but its statistical efficiency can be improved by exploiting the structures of the utility function. Adapting the Regression MSR method Witter et al. (2026) using a quadratic proxy Fumagalli et al. (2026) to our priority-aware setting, we develop a proxy-plus-bias-correction estimator. Concretely, we fit a quadratic proxy

$$\widehat{U}(S) = U(\emptyset) + \sum_{i \in S} a_i + \sum_{\{i,j\} \subseteq S, i \parallel j} b_{ij}, \quad (10)$$

where $i \parallel j$ denotes that neither $i \preceq j$ nor $j \preceq i$. We compute PASV by

$$\psi(U) = \psi(\widehat{U}) + \psi(U - \widehat{U}), \quad (11)$$

in which the first term $\psi(\widehat{U})$ can be estimated in a closed form by using fitted $a$ and $b$ values, leaving only the second term (i.e. residual) $\psi(U - \widehat{U})$ to be estimated by Monte Carlo. Whether the proxy improves over the direct Monte Carlo estimator depends on the number of players $n$ and on how much low-order structure $U$ exhibits. Appendix D.2 reports a head-to-head comparison of these methods up to $n = 2048$: the proxy reduces approximation error under a fixed budget when $n$ is moderate (e.g. $n \leq 512$), but the proxy-fit cost scales as $O(N_{\text{coef}} d^2)$ with $d$ up to $O(n^2)$ and

the peak memory grows accordingly (tens of GB at $n = 2048$; Figure 12). Choosing between the direct estimator and the proxy variant is therefore closely tied to the balance between proxy quality and proxy budget.

## 6.2. Sensitivity to DAG Change

Since the DAG is a modeling input rather than an observed ground truth, a natural deployment concern is how PASV's output depends on the chosen DAG. A crucial point is that the right answer here is not blanket robustness. For example, suppose $i$ and $j$ both claim a major discovery; then the three cases: $i \prec j$, $j \prec i$, and $i \parallel j$ (independent discovery) are *expected to produce remarkably different PASV results* for $i$ and $j$. Therefore, we provide practitioners not a simple claim of robustness against DAG changes, but instead a diagnostic tool for the DAG-sensitivity of the result, similar in spirit to that of priority sweeping for $\lambda$. By varying one edge at a time and tracking the change in PASV, users obtain a per-edge importance ranking that flags which edges of their DAG warrant careful verification and which can be altered without significant consequence.

We revisit the feature-attribution setting of Section 5.2, fix $\lambda \equiv 1$, and perturb the general DAG of Figure 6b under four scenarios: (i) single-edge deletion (12 transitive-reduction edges); (ii) single-edge addition (20 sampled valid additions); (iii) minimum-cut deletions disconnecting the DAG into two equal components; and (iv) deleting or keeping only the top-$K$ most influential edges. Let $\psi$ and $\psi'$ be the PASV vectors under original and altered DAG's. We measure the Relative Attribution Shift $\text{RAS} = \|\psi' - \psi\|_2 / \|\psi\|_2$ and pair it with auxiliary diagnostics including the Pearson correlation of $\psi$ and $\psi'$, the top-4 overlap, and the pairwise-order change $\Delta_{\text{ord}}$. Appendix F.4.2 reports the full results (Figure 19). The behavior matches the intended use of our diagnostic tool: most single-edge perturbations have small RAS (below 0.05) with Pearson correlation above 0.97, while a few critical edges and disconnecting cuts produce visibly larger shifts; across all scenarios RAS is closely tied to $\Delta_{\text{ord}}$. These importance scores provide users a clean priority list for DAG edge validation.

## 7. Conclusion

We introduce Priority-Aware Shapley Value (PASV), a unified Shapley-based attribution method that simultaneously enforces hard precedence constraints and incorporates soft priority weights. PASV recovers precedence-only and weight-only Shapley variants as special cases, admits scalable MCMC estimation for general DAGs, and has a solid axiomatization basis. An important tool that PASV provides practitioners is a sensitivity/robustness diagnostic procedure via *priority sweeping*.

## Impact Statement

This paper presents work whose goal is to advance the field of Machine Learning. There are many potential societal consequences of our work, none of which we feel must be specifically highlighted here.

## Acknowledgement

Lee and Zhang were supported by NSF DMS-2311109. Liu and Tang were supported by NSF DMS-2412853 and Jane Street Group, LLC.

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

## A. Standard Axioms of the Shapley Value

We briefly review the standard axiomatic framework leading to the Shapley value. Throughout this appendix we fix the player set $[n] = \{1, \ldots, n\}$ and assume $U(\emptyset) = 0$ without loss of generality, since replacing $U$ by $U'(S) := U(S) - U(\emptyset)$ leaves every marginal contribution $U(S \cup \{i\}) - U(S)$ unchanged. A *value* is a mapping $\psi$ that assigns to each utility function $U$ a payoff vector $\psi(U) = (\psi_i(U))_{i \in [n]}$. The Shapley value $\psi^{\text{SV}}$ is characterized as the unique value satisfying the following four standard axioms (Shapley, 1953).

**Axiom A.1** (Efficiency; E)**.** For a given utility function $U$,

$$\sum_{i \in [n]} \psi_i(U) = U([n]).$$

**Axiom A.2** (Linearity; L)**.** For given two utility functions $U, V$ and all $\alpha, \beta \in \mathbb{R}$,

$$\psi(\alpha U + \beta V) = \alpha \psi(U) + \beta \psi(V).$$

**Axiom A.3** (Null Player; NP)**.** If player $i$ satisfies $U(S \cup \{i\}) = U(S)$ for all $S \subseteq [n] \setminus \{i\}$, then

$$\psi_i(U) = 0.$$

**Axiom A.4** (Symmetry; S)**.** If two players $i$ and $j$ satisfy $U(S \cup \{i\}) = U(S \cup \{j\})$ for all $S \subseteq [n] \setminus \{i, j\}$, then

$$\psi_i(U) = \psi_j(U).$$

Efficiency requires that the entire value $U([n])$ be fully allocated among players. Linearity ensures that payoffs respect additive changes in utility. That is, if a game is formed by combining (possibly with weights) two utilities, then the assigned payoffs combine in the same way. Null player formalizes a fairness requirement in the sense of that a player who never changes the utility receives zero payoff. Symmetry reflects anonymity by requiring identical treatment of players who are interchangeable in terms of their marginal contributions.

## B. Proofs of Propositions on Limiting Cases in Section 3.4

### B.1. Proof of Proposition 3.13

*Proof.* Fix $i \in \max_{\preceq}([n])$ and fix $(\lambda_j)_{j \neq i}$. Let $([n], \preceq')$ and $\lambda'$ be the modified poset and weight vector satisfying the properties in the statement, respectively. Since $p^{(\preceq, \lambda)}$ is defined in proportional form, it is convenient to work with the corresponding unnormalized weight

$$w^{(\preceq, \lambda)}(\pi) := \prod_{t=1}^{n} \frac{|\max_{\preceq}(S_t)| \lambda_{\pi_t}}{\sum_{k \in \max_{\preceq}(S_t)} \lambda_k}.$$

Since $\preceq'$ adds precedence relations to $\preceq$, every $\pi$ that respects $\preceq'$ also respects $\preceq$, and hence $\Pi^{\preceq'} \subseteq \Pi^{\preceq}$. We consider separately the cases $\pi \in \Pi^{\preceq'}$ and $\pi \in \Pi^{\preceq} \setminus \Pi^{\preceq'}$.

First, suppose $\pi \in \Pi^{\preceq'}$. In this case, the added relations $j \prec i$ for all $j \in \max_{\preceq}([n]) \setminus \{i\}$ force $i$ to be the last element in $\pi$, i.e., $\pi_n = i$. Hence we can write

$$
\begin{aligned}
w^{(\preceq, \lambda)}(\pi) &= \left( \prod_{t=1}^{n-1} \frac{|\max_{\preceq}(S_t)| \lambda_{\pi_t}}{\sum_{k \in \max_{\preceq}(S_t)} \lambda_k} \right) \cdot \left( \frac{|\max_{\preceq}([n])| \lambda_i}{\sum_{k \in \max_{\preceq}([n])} \lambda_k} \right) \\
&\to \prod_{t=1}^{n-1} \frac{|\max_{\preceq}(S_t)| \lambda_{\pi_t}}{\sum_{k \in \max_{\preceq}(S_t)} \lambda_k},
\end{aligned}
\tag{12}
$$

as $\lambda_i \to \infty$, since $i \in \max_{\preceq}([n])$ implies

$$\frac{|\max_{\preceq}([n])| \lambda_i}{\sum_{k \in \max_{\preceq}([n])} \lambda_k} = \frac{|\max_{\preceq}([n])| \lambda_i}{\lambda_i + \sum_{k \in \max_{\preceq}([n]) \setminus \{i\}} \lambda_k} \to 1.$$

On the other hand, the unnormalized weight for $p^{(\preceq', \lambda')}$ is

$$w^{(\preceq', \lambda')}(\pi) = \left( \prod_{t=1}^{n-1} \frac{|\max_{\preceq'}(S_t)| \lambda'_{\pi_t}}{\sum_{k \in \max_{\preceq'}(S_t)} \lambda'_k} \right) \cdot \left( \frac{|\max_{\preceq'}([n])| \lambda'_i}{\sum_{k \in \max_{\preceq'}([n])} \lambda'_k} \right). \tag{13}$$

Because $\pi_n = i$, for every $t \in \{1, \ldots, n-1\}$ the remaining set $S_t$ does not contain $i$. By construction of $\preceq'$, the only new relations are of the form $j \prec i$ for $j \in \max_{\preceq}([n]) \setminus \{i\}$, which do not affect the admissible sets once $i$ is removed. Therefore, for $t \in [n-1]$,

$$\max_{\preceq'}(S_t) = \max_{\preceq}(S_t).$$

Moreover, by the definition of $\lambda'$ we have $\lambda'_j = \lambda_j$ for all $j \neq i$, and since $\pi_t \neq i$ for $t \in [n-1]$, it follows that $\lambda'_{\pi_t} = \lambda_{\pi_t}$ for all $t \in [n-1]$. Hence the first product term in (13) coincides with the limit in (12).

Finally, under $\preceq'$ the element $i$ is the unique maximal element of $[n]$, i.e., $\max_{\preceq'}([n]) = \{i\}$. Thus the second factor in (13) equals 1. Combining these observations, for $\pi \in \Pi^{\preceq'}$, we obtain

$$\lim_{\lambda_i \to \infty} w^{(\preceq, \lambda)}(\pi) = w^{(\preceq', \lambda')}(\pi). \tag{14}$$

Next, consider the case that $\pi \in \Pi^{\preceq} \setminus \Pi^{\preceq'}$. In this case, $\pi$ respects $\preceq$ but violates at least one of the additional relations defining $\preceq'$. Let $t^\star$ be the step such that $\pi_{t^\star} = j$. Then $i \in S_{t^\star}$ since $i$ has not been removed before step $t^\star$. Moreover, since $i, j \in \max_{\preceq}([n])$, both are maximal whenever they are present, and hence

$$\{i, j\} \subseteq \max_{\preceq}(S_{t^\star}).$$

Therefore, the $t^\star$-th factor in $w^{(\preceq, \lambda)}(\pi)$ satisfies

$$\frac{|\max_{\preceq}(S_{t^\star})| \lambda_j}{\sum_{k \in \max_{\preceq}(S_{t^\star})} \lambda_k} \leq \frac{|\max_{\preceq}(S_{t^\star})| \lambda_j}{\lambda_i + \lambda_j}.$$

All other factors in the product defining $w^{(\preceq, \lambda)}(\pi)$ are bounded above by $|\max_{\preceq}(S_t)|$ and do not diverge with $\lambda_i$. Hence there exists a constant $C(\pi) < \infty$ which is independent of $\lambda_i$ such that

$$w^{(\preceq, \lambda)}(\pi) \leq C(\pi) \cdot \frac{\lambda_j}{\lambda_i + \lambda_j} \to 0. \tag{15}$$

as $\lambda_i \to \infty$.

Let $Z(\lambda_i) := \sum_{\pi \in \Pi^{\preceq}} w^{(\preceq, \lambda)}(\pi)$ and $Z' := \sum_{\pi \in \Pi^{\preceq'}} w^{(\preceq', \lambda')}(\pi)$. Since $\Pi^{\preceq}$ is finite, by (14) and (15) we have

$$\begin{aligned} Z(\lambda_i) &= \sum_{\pi \in \Pi^{\preceq'}} w^{(\preceq, \lambda)}(\pi) + \sum_{\pi \in \Pi^{\preceq} \setminus \Pi^{\preceq'}} w^{(\preceq, \lambda)}(\pi) \\ &\to \sum_{\pi \in \Pi^{\preceq'}} w^{(\preceq', \lambda')}(\pi) + 0 \\ &=: Z' > 0. \end{aligned}$$

Therefore, for any $\pi \in \Pi^{\preceq'}$,

$$p^{(\preceq, \lambda)}(\pi) = \frac{w^{(\preceq, \lambda)}(\pi)}{Z(\lambda_i)} \to \frac{w^{(\preceq', \lambda')}(\pi)}{Z'} = p^{(\preceq', \lambda')}(\pi),$$

while for any $\pi \in \Pi^{\preceq} \setminus \Pi^{\preceq'}$,

$$p^{(\preceq, \lambda)}(\pi) = \frac{w^{(\preceq, \lambda)}(\pi)}{Z(\lambda_i)} \to 0,$$

as $\lambda_i \to \infty$. This completes the proof.

$\square$

## B.2. Proof of Proposition 3.14

To prove Proposition 3.14, we first introduce two useful lemmas. Lemma B.1 shows that, under an ordered-partition precedence relation, the induced order distribution factorizes into the product of within-block distributions. This factorization allows us to localize the limiting argument to the single block $B$ in which the weights are rescaled, while the contributions from all other blocks remain unchanged. Lemma B.2 establishes the basic limiting mechanism in the absence of precedence constraints. In particular, sending the weights of a subset $G$ to infinity forces $G$ to form a late block in the induced order distribution. The proofs of Lemmas B.1 and B.2 are given in Appendices B.2.1 and B.2.2, respectively.

**Lemma B.1** (Blockwise factorization under ordered partitions)**.** *Suppose a poset* $([n], \preceq)$ *is induced by an ordered partition* $(B_1, \ldots, B_m)$. *Let* $\pi \in \Pi^{\preceq}$, *and for each* $\ell \in [m]$ *let* $\pi_{B_\ell}$ *denote the relative order of players in* $B_\ell$ *induced by* $\pi$. *Let* $\lambda_{B_\ell}$ *be the restriction of* $\lambda$ *to* $B_\ell$, *and let* $\preceq_{B_\ell}$ *denote the trivial partial order on* $B_\ell$. *Then the priority-aware distribution* $p^{(\preceq,\lambda)}$ *admits the factorization*

$$p^{(\preceq,\lambda)}(\pi) = \prod_{\ell=1}^{m} p^{(\preceq_{B_\ell}, \lambda_{B_\ell})}(\pi_{B_\ell}), \tag{16}$$

*where each factor on the right-hand side is the priority-aware distribution on the player set* $B_\ell$ *under the trivial precedence relation.*

**Lemma B.2.** *Suppose a poset* $([n], \preceq)$ *is trivial in the sense that there is no pair* $i, j \in [n]$ *such that* $i \neq j$ *and* $i \prec j$, *i.e. no edge in an induced DAG. Fix a nonempty subset* $G \subseteq [n]$ *and fix weights* $(\lambda_j)_{j \notin G}$. *For* $i \in G$ *set* $\lambda_i = \bar{\lambda}$ *and let* $\bar{\lambda} \to \infty$. *Let* $\preceq'$ *be the partial order induced by the ordered partition* $([n] \setminus G, G)$. *Then, for any* $\lambda'$ *satisfying* $\lambda'_j = \lambda_j$ *for all* $j \notin G$, $\lambda'_i = \widetilde{\lambda}$ *for* $i \in G$ *with arbitrary* $\widetilde{\lambda} > 0$, *and for every* $\pi \in \Pi$,

$$\lim_{\bar{\lambda} \to \infty} p^{(\preceq,\lambda)}(\pi) = \begin{cases} p^{(\preceq',\lambda')}(\pi) & \text{if } \pi \in \Pi^{\preceq'}, \\ 0 & \text{otherwise.} \end{cases}$$

Now we provide the proof of Proposition 3.14.

*Proof of Proposition 3.14.* By Lemma B.1, for any $\pi \in \Pi^{\preceq}$,

$$p^{(\preceq,\lambda)}(\pi) = \prod_{\ell=1}^{m} p^{(\preceq_{B_\ell}, \lambda_{B_\ell})}(\pi_{B_\ell}).$$

All factors with $B_\ell \neq B$ are constant in $\bar{\lambda}$, so it suffices to analyze the single factor on $B$.

Applying Lemma B.2 to the block-$B$ factor yields

$$\lim_{\bar{\lambda} \to \infty} p^{(\preceq_B, \lambda_B)}(\pi_B) = \begin{cases} p^{(\preceq'_B, \lambda'_B)}(\pi_B) & \text{if } \pi_B \in \Pi^{\preceq'_B}, \\ 0 & \text{otherwise,} \end{cases}$$

with $\preceq'_B$ induced by $(B \setminus G, G)$ and $\lambda'_B$ agreeing with $\lambda_B$ on $B \setminus G$ while taking an arbitrary common positive weight on $G$.

Now observe that $\pi \in \Pi^{\preceq'}$ holds if and only if $\pi \in \Pi^{\preceq}$ and $\pi_B \in \Pi^{\preceq'_B}$. Recalling Lemma B.1 again for $(\preceq', \lambda')$ and combining it with the previous results gives, for every $\pi \in \Pi$,

$$\lim_{\bar{\lambda} \to \infty} p^{(\preceq,\lambda)}(\pi) = \begin{cases} p^{(\preceq',\lambda')}(\pi) & \text{if } \pi \in \Pi^{\preceq'}, \\ 0 & \text{otherwise,} \end{cases}$$

which completes the proof. $\square$

### B.2.1. PROOF OF LEMMA B.1

*Proof.* By Proposition 3.2, for $\pi \in \Pi^{\preceq}$,

$$p^{(\preceq,\lambda)}(\pi) = \prod_{t=1}^{n} \frac{\lambda_{\pi_t}}{\sum_{k \in \max_{\preceq}(S_t)} \lambda_k}. \tag{17}$$

For each $\ell \in [m]$, define
$$I_\ell(\pi) := \{t \in [n] : \pi_t \in B_\ell\}.$$

Since $\pi$ respects the ordered-partition precedence, each $I_\ell(\pi)$ is a contiguous interval and $\{I_\ell(\pi)\}_{\ell=1}^m$ partitions $[n]$.

Fix $\ell \in [m]$ and take any $t \in I_\ell(\pi)$. Then $S_t$ contains no elements from blocks later than $B_\ell$ and satisfies $S_t \cap B_\ell \neq \emptyset$. Hence the admissible set at step $t$ is exactly the remaining players in $B_\ell$:
$$\max_{\preceq}(S_t) = S_t \cap B_\ell, \quad t \in I_\ell(\pi). \tag{18}$$

Substituting (18) into (17) and regrouping terms gives
$$p^{(\preceq,\lambda)}(\pi) = \prod_{\ell=1}^m \prod_{t \in I_\ell(\pi)} \frac{\lambda_{\pi_t}}{\sum_{k \in S_t \cap B_\ell} \lambda_k}. \tag{19}$$

It remains to identify each product inside with the priority-aware distribution on the block $B_\ell$ under trivial precedence. By Proposition 3.2 applied to $(B_\ell, \preceq_{B_\ell})$, the corresponding distribution satisfies, for $\rho \in \Pi^{\preceq_{B_\ell}}$,
$$p^{(\preceq_{B_\ell}, \lambda_{B_\ell})}(\rho) = \prod_{r=1}^{|B_\ell|} \frac{(\lambda_{B_\ell})_{\rho_r}}{\sum_{k \in T_r} (\lambda_{B_\ell})_k}, \tag{20}$$

where $T_r = \{\rho_1, \ldots, \rho_r\}$. Taking $\rho = \pi_{B_\ell}$ and matching $T_r$ with $S_t \cap B_\ell$ along $t \in I_\ell(\pi)$ yields
$$\prod_{t \in I_\ell(\pi)} \frac{\lambda_{\pi_t}}{\sum_{k \in S_t \cap B_\ell} \lambda_k} = p^{(\preceq_{B_\ell}, \lambda_{B_\ell})}(\pi_{B_\ell}).$$

Plugging this into (19) proves the claim. $\qquad\square$

### B.2.2. PROOF OF LEMMA B.2

*Proof.* Since the poset $([n], \preceq)$ is trivial, $\Pi^{\preceq} = \Pi$ and $\max_{\preceq}(S) = S$ for all $S \subseteq [n]$. Furthermore, since a trivial poset is also an ordered partition, by Proposition 3.2, the priority-aware distribution coincides with $p_{WSV}$ in (3). Hence, the distribution takes the form
$$p^{(\preceq,\lambda)}(\pi) = \prod_{t=1}^n \frac{\lambda_{\pi_t}}{\sum_{k \in S_t} \lambda_k}. \tag{21}$$

On the other hand, under the modified poset $([n], \preceq')$ induced by the ordered partition $([n] \setminus G, G)$, the admissible set is
$$\max_{\preceq'}(S) = \begin{cases} S \cap G, & \text{if } S \cap G \neq \emptyset, \\ S, & \text{if } S \cap G = \emptyset, \end{cases}$$

since all elements of $[n] \setminus G$ precede all elements of $G$. Therefore, again by Proposition 3.2, the corresponding priority-aware distribution satisfies
$$
\begin{aligned}
p^{(\preceq',\lambda')}(\pi) &= \prod_{t=1}^n \frac{\lambda'_{\pi_t}}{\sum_{k \in \max_{\preceq'}(S_t)} \lambda'_k} \\
&= \left( \prod_{t=1}^{n-|G|} \frac{\lambda'_{\pi_t}}{\sum_{k \in S_t} \lambda'_k} \right) \cdot \left( \prod_{t=n-|G|+1}^n \frac{\lambda'_{\pi_t}}{\sum_{k \in S_t \cap G} \lambda'_k} \right) \\
&= \left( \prod_{t=1}^{n-|G|} \frac{\lambda'_{\pi_t}}{\sum_{k \in S_t} \lambda'_k} \right) \cdot \left( \prod_{t=n-|G|+1}^n \frac{1}{|S_t \cap G|} \right)
\end{aligned} \tag{22}
$$

for $\pi \in \Pi^{\preceq'}$. The last line follows from $\lambda'_i = \widetilde{\lambda}$ for $i \in G$.

Now consider the case of $\pi \in \Pi^{\preceq'}$. Then since all elements of $[n] \setminus G$ appear before all elements of $G$ in $\pi$, we also can decompose the probability from the original poset in (21):

$$p^{(\preceq,\lambda)}(\pi) = \left( \prod_{t=1}^{n-|G|} \frac{\lambda_{\pi_t}}{\sum_{k \in S_t} \lambda_k} \right) \cdot \left( \prod_{t=n-|G|+1}^{n} \frac{\lambda_{\pi_t}}{\sum_{k \in S_t} \lambda_k} \right), \tag{23}$$

In addition, at every step $t \in \{n - |G| + 1, \ldots, n\}$, we have $\lambda_{\pi_t} = \bar{\lambda}$ and $S_t \cap G \neq \emptyset$. This yields

$$\frac{\lambda_{\pi_t}}{\sum_{k \in S_t} \lambda_k} = \frac{\bar{\lambda}}{|S_t \cap G|\bar{\lambda} + \sum_{k \in S_t \setminus G} \lambda_k} \to \frac{1}{|S_t \cap G|},$$

as $\bar{\lambda} \to \infty$, which makes the limit of the second term in (23) coincide with the second term in (22). Furthermore, since $\lambda_i' = \lambda_i$ for $i \in [n] \setminus G$, this proves

$$\lim_{\bar{\lambda} \to \infty} p^{(\preceq,\lambda)}(\pi) = p^{(\preceq',\lambda')}(\pi),$$

for $\pi \in \Pi^{\preceq'}$.

Next, suppose that $\pi \notin \Pi^{\preceq'}$. Then there exist $i \in G$ and $j \in [n] \setminus G$ such that $i$ appears before $j$ in $\pi$. Let $t^\star$ be a step such that $\pi_{t^\star} = j$ and $i \in S_{t^\star}$. Then the $t^\star$-th factor satisfies

$$\frac{\lambda_j}{\sum_{k \in S_{t^\star}} \lambda_k} \leq \frac{\lambda_j}{\lambda_i} = \frac{\lambda_j}{\bar{\lambda}} \to 0,$$

as $\bar{\lambda} \to \infty$, since $\lambda_i = \bar{\lambda}$. All other factors are bounded above by 1, hence $p^{(\preceq,\lambda)}(\pi) \to 0$ as $\bar{\lambda} \to \infty$. $\square$

## C. Proofs of Main Theoretical Results

### C.1. Proof of Theorem 3.8

We first recall a classical characterization of random order values. The following lemma is useful for the main proof.

**Lemma C.1** (Axiomatization of ROV). *A value $\psi$ is ROV if and only if it satisfies axioms E + L + NP + M.*

We defer the proof of the lemma above to standard references (See Theorem 2 in Derks (2005)). The proof of Theorem 3.8 is as follows:

*Proof.* Assume $\psi$ satisfies E + L + NP + M + MS. By Lemma C.1, $\psi$ is an ROV, i.e., there exists a distribution $p$ on $\Pi$ such that (1) holds. It therefore suffices to show that $p(\pi) = 0$ for all $\pi \in \Pi \setminus \Pi^{\preceq}$ if and only if $\psi$ satisfies MS.

First, we will show the forward direction. Assume that $p(\pi) = 0$ for all $\pi \in \Pi \setminus \Pi^{\preceq}$. Choose any feasible set $T \in \mathcal{S}^{\preceq}$ and consider the corresponding elementary game $u_T$. For each $i \in [n]$, using (1) and the definition of $u_T$, we can show

$$\begin{aligned}
\psi_i(u_T) &= \mathbb{E}_{\pi \sim p} \left[ u_T(\pi^i \cup \{i\}) - u_T(\pi^i) \right] \\
&= \mathbb{E}_{\pi \sim p} \left[ \mathbb{1}\{T \not\subseteq \pi^i, T \subseteq \pi^i \cup \{i\}\} \right] \\
&= \mathbb{P}_{\pi \sim p} \left( i \text{ is last for } T \text{ in } \pi \right) \\
&= \sum_{\{\pi : i \text{ is last for } T \text{ in } \pi\}} p(\pi). \tag{24}
\end{aligned}$$

In particular, if $i \notin \max_{\preceq}(T)$, then no linear extension $\pi \in \Pi^{\preceq}$ can have $i$ as the last element of $T$. Since $p$ is supported on $\Pi^{\preceq}$, the sum in (24) is zero and hence $\psi_i(u_T) = 0$. This verifies MS.

Conversely, suppose there exists $\pi^\star \in \Pi \setminus \Pi^{\preceq}$ with $p(\pi^\star) > 0$. Then $\pi^\star$ violates the precedence relation, so there exist a violating pair of players $i, j \in [n]$ with $i \prec j$ but $\pi^\star(j) < \pi^\star(i)$. Let $\mathcal{P}(j) := \{i \in [n] : i \prec j\}$ denote the predecessor set of $j$ under $\preceq$ and let $k \in \mathcal{P}(j)$ be a predecessor that appears latest among $\mathcal{P}(j)$ in $\pi^\star$. Define

$$T := \mathcal{P}(j) \cup \{j\}.$$

By construction, $T$ is feasible, and $k \notin \max_{\preceq}(T)$ since $k \prec j \in T$. Moreover, under $\pi^{\star}$, every element of $T \setminus \{k\}$ appears before $k$, so $k$ is last for $T$ in $\pi^{\star}$. Therefore, (24) implies

$$\psi_k(u_T) \geq p(\pi^{\star}) > 0,$$

which contradicts MS. Hence no such $\pi^{\star}$ can have positive probability, and this completes the proof.

$\square$

### C.2. Proof of Theorem 3.12

*Proof.* Since $p$ is PROV, it is supported on $\Pi^{\preceq}$. Assume $p$ admits the SCF form (6):

$$p(\pi) \propto \prod_{t=1}^{n} s_{\preceq}(S_t) c_{\preceq,\lambda}(\pi_t; S_t),$$

for $\pi \in \Pi^{\preceq}$, where $\sum_{i \in \max_{\preceq}(S_t)} c_{\preceq,\lambda}(i; S_t) = 1$.

First, by WP, for every feasible set $S$ and every $i, j \in \max_{\preceq}(S)$,

$$\frac{c_{\preceq,\lambda}(i; S)}{c_{\preceq,\lambda}(j; S)} = \frac{\lambda_i}{\lambda_j}.$$

Together with $\sum_{i \in \max_{\preceq}(S)} c_{\preceq,\lambda}(i; S) = 1$, for $i \in \max_{\preceq}(S)$, this implies

$$c_{\preceq,\lambda}(i; S) = \frac{\lambda_i}{\sum_{k \in \max_{\preceq}(S)} \lambda_k}. \tag{25}$$

Now, without loss of generality, assume $\lambda_i = 1$ for all $i \in [n]$. Then (25) gives $c_{\preceq,\lambda}(i; S) = 1/|\max_{\preceq}(S)|$, so for $\pi \in \Pi^{\preceq}$,

$$p(\pi) \propto \prod_{t=1}^{n} \frac{s_{\preceq}(S_t)}{|\max_{\preceq}(S_t)|}. \tag{26}$$

By EWU, $p$ must be uniform on $\Pi^{\preceq}$ under constant weights. Hence the product in (26) is a constant independent of $\pi \in \Pi^{\preceq}$:

$$\prod_{t=1}^{n} \frac{s_{\preceq}(S_t)}{|\max_{\preceq}(S_t)|} = K, \tag{27}$$

for all $\pi \in \Pi^{\preceq}$.

For general $\lambda$, combining SCF with (25) yields, for $\pi \in \Pi^{\preceq}$,

$$\begin{aligned} p(\pi) &\propto \prod_{t=1}^{n} s_{\preceq}(S_t) \frac{\lambda_{\pi_t}}{\sum_{k \in \max_{\preceq}(S_t)} \lambda_k} \\ &= \left( \prod_{t=1}^{n} \frac{s_{\preceq}(S_t)}{|\max_{\preceq}(S_t)|} \right) \cdot \left( \prod_{t=1}^{n} \frac{\lambda_{\pi_t} |\max_{\preceq}(S_t)|}{\sum_{k \in \max_{\preceq}(S_t)} \lambda_k} \right) \\ &\propto \prod_{t=1}^{n} \frac{\lambda_{\pi_t} |\max_{\preceq}(S_t)|}{\sum_{k \in \max_{\preceq}(S_t)} \lambda_k}. \end{aligned}$$

The last proportionality is due to (27), and this proves that $p$ is PASV.

Conversely, the PASV distribution (4) is realized in the SCF class by taking $c_{\preceq,\lambda}(i; S) = \lambda_i / \sum_{k \in \max_{\preceq}(S)} \lambda_k$ and an appropriate state factor, so WP holds by construction. If $\lambda_i = 1$ for $i \in [n]$, each factor in (4) reduces to 1, so PASV is uniform on $\Pi^{\preceq}$ and EWU holds. $\square$

### C.3. Proof of Proposition 3.1

*Proof.* Without loss of generality, assume $\lambda_i = 1$ for $i \in [n]$. For any $\pi \in \Pi^{\preceq}$, the PASV weight (4) satisfies

$$p^{(\preceq,\lambda)}(\pi) \;\propto\; \prod_{t=1}^{n} \frac{\lambda_{\pi_t} |\max_{\preceq}(S_t)|}{\sum_{k \in \max_{\preceq}(S_t)} \lambda_k} = \prod_{t=1}^{n} \frac{|\max_{\preceq}(S_t)|}{|\max_{\preceq}(S_t)|} = 1.$$

Hence $p^{(\preceq,\lambda)}$ is uniform over $\Pi^{\preceq}$, which coincides with $p_{\text{PSV}}$. $\qquad\square$

### C.4. Proof of Proposition 3.2

*Proof.* Assume $\preceq$ is induced by an ordered partition $(B_1, \ldots, B_m)$. Fix any $\pi \in \Pi^{\preceq}$ and let $S_t = \{\pi_1, \ldots, \pi_t\}$. Under an ordered partition, at each step the admissible set $\max_{\preceq}(S_t)$ consists exactly of the players in the earliest block that still has remaining elements. In particular, if at some step $t$ the current active block has $r$ remaining elements, then $|\max_{\preceq}(S_t)| = r$, regardless of which specific element is chosen within that block. Therefore,

$$\prod_{t=1}^{n} |\max_{\preceq}(S_t)| = \prod_{\ell=1}^{m} |B_\ell|!,$$

which is a constant independent of $\pi \in \Pi^{\preceq}$.

Using (4), for any $\pi \in \Pi^{\preceq}$ we can write

$$
\begin{aligned}
p^{(\preceq,\lambda)}(\pi) &\propto \prod_{t=1}^{n} \frac{\lambda_{\pi_t} |\max_{\preceq}(S_t)|}{\sum_{k \in \max_{\preceq}(S_t)} \lambda_k} \\
&= \left( \prod_{t=1}^{n} |\max_{\preceq}(S_t)| \right) \cdot \left( \prod_{t=1}^{n} \frac{\lambda_{\pi_t}}{\sum_{k \in \max_{\preceq}(S_t)} \lambda_k} \right) \\
&\propto \prod_{t=1}^{n} \frac{\lambda_{\pi_t}}{\sum_{k \in \max_{\preceq}(S_t)} \lambda_k},
\end{aligned}
$$

where the last proportionality uses that $\prod_{t=1}^{n} |\max_{\preceq}(S_t)|$ is constant over $\Pi^{\preceq}$. The remaining product is exactly the weighted Shapley distribution $p_{\text{WSV}}(\pi)$ under the ordered-partition precedence. Hence $p^{(\preceq,\lambda)}(\pi) = p_{\text{WSV}}(\pi)$ for all $\pi \in \Pi^{\preceq}$. $\qquad\square$

### C.5. Proof of Lemma 4.1

*Proof.* Fixing a poset $([n], \preceq)$ and weights $\lambda$, for a given $\pi \in \Pi^{\preceq}$ let $\pi' \in \Pi^{\preceq}$ be obtained by swapping an incomparable pair $(\pi_k, \pi_{k+1})$ in $\pi$. Then, the probability ratio of $\pi$ and $\pi'$ under $p^{(\preceq,\lambda)}$ is given by:

$$\frac{p^{(\preceq,\lambda)}(\pi')}{p^{(\preceq,\lambda)}(\pi)} = \prod_{t=1}^{n} \left[ \frac{\lambda_{\pi'_t} |\max_{\preceq}(S'_t)|}{\sum_{j \in \max_{\preceq}(S'_t)} \lambda_j} \bigg/ \frac{\lambda_{\pi_t} |\max_{\preceq}(S_t)|}{\sum_{i \in \max_{\preceq}(S_t)} \lambda_i} \right],$$

where $S'_t = \{\pi'_1, \ldots, \pi'_t\}$. Then, since $\pi'_t = \pi_t$ for all $t \notin \{k, k+1\}$ and $S'_t = S_t$ for $t \neq k$, it remains to compare the factors only at $t = k$ and $t = k+1$. For $t = k+1$, we have $S'_{k+1} = S_{k+1}$, and hence $\max_{\preceq}(S'_{k+1}) = \max_{\preceq}(S_{k+1})$, which gives

$$\frac{\lambda_{\pi'_{k+1}} |\max_{\preceq}(S'_{k+1})|}{\sum_{j \in \max_{\preceq}(S'_{k+1})} \lambda_j} \bigg/ \frac{\lambda_{\pi_{k+1}} |\max_{\preceq}(S_{k+1})|}{\sum_{i \in \max_{\preceq}(S_{k+1})} \lambda_i} = \frac{\lambda_{\pi_k}}{\lambda_{\pi_{k+1}}}.$$

For the $t = k$ factor, using $\pi'_k = \pi_{k+1}$ gives

$$\frac{\lambda_{\pi'_k} |\max_{\preceq}(S'_k)|}{\sum_{j \in \max_{\preceq}(S'_k)} \lambda_j} \bigg/ \frac{\lambda_{\pi_k} |\max_{\preceq}(S_k)|}{\sum_{i \in \max_{\preceq}(S_k)} \lambda_i} = \frac{\lambda_{\pi_{k+1}}}{\lambda_{\pi_k}} \cdot \frac{|\max_{\preceq}(S'_k)|}{|\max_{\preceq}(S_k)|} \cdot \frac{\sum_{i \in \max_{\preceq}(S_k)} \lambda_i}{\sum_{j \in \max_{\preceq}(S'_k)} \lambda_j}.$$

Multiplying the two ratios above cancels the weight ratio of $\pi_k$ and $\pi_{k+1}$, yielding

$$\frac{p^{(\preceq,\lambda)}(\pi')}{p^{(\preceq,\lambda)}(\pi)} = \frac{\left(\sum_{i\in\max_\preceq(S_k)}\lambda_i\right)/|\max_\preceq(S_k)|}{\left(\sum_{j\in\max_\preceq(S'_k)}\lambda_j\right)/|\max_\preceq(S'_k)|},$$

which proves the claim. □

# D. Simulation Studies

The experiments in Section 5 demonstrate PASV in two applied settings, where the utility evaluation itself is tied to a downstream learning task. In this section, we complement those experiments with controlled simulation studies that isolate the computational and statistical behavior of the PASV estimators. The first set of simulations studies the direct Monte Carlo estimator, separating the behavior of the MCMC sampler from the accuracy and cost of the retained samples. The second set studies a proxy-based variant that uses a quadratic regression approximation to reduce Monte Carlo error under the same total sampling budget.

## D.1. Estimation Accuracy

The basic PASV estimator has two stages. The burn-in stage runs the adjacent-swap M-H chain until its distribution is close to the target distribution $p^{(\preceq,\lambda)}$, but does not evaluate the utility. The Monte Carlo stage keeps thinned samples from the chain and averages the marginal contributions induced by those sampled linear extensions. Accordingly, we first examine how quickly the chain mixes, and then evaluate how the retained sample size affects the accuracy and computational cost of PASV estimation.

### D.1.1. MIXING TIME ANALYSIS OF MCMC SAMPLER

We first study the mixing behavior of the adjacent-swap M-H sampler. In this experiment, we investigate whether the sampler itself becomes a computational bottleneck when the target distribution is no longer uniform over linear extensions because of non-uniform priority weights. For a sampled linear extension $\pi$, write $i \prec_\pi j$ if player $i$ appears before player $j$ in $\pi$. Let $P^t_{\pi_0}$ denote the distribution of the chain after $t$ M-H iterations initialized at $\pi_0$, and let $P^\star$ denote the stationary distribution. We measure mixing through the pairwise-order discrepancy

$$D_t = \max_{i,j\in[n]}\left|\mathbb{P}^t_{\pi_0}(i\prec_\pi j) - \mathbb{P}^\star(i\prec_\pi j)\right|. \tag{28}$$

The empirical mixing time is defined as the earliest $t$ such that $D_t < 1/4$.

We use four synthetic poset families from Talvitie et al. (2017), each with two sub-settings: AveDeg with parameters 2 and 4, MaxInDeg with parameters 2 and 4, GridTree with parameters 2 and 4, and Bipartite with parameters 0.2 and 0.5. The AveDeg($k$) family adds each forward arc $a \to b$ with $a < b$ independently with probability $k/(n-1)$, yielding sparse random DAGs with controlled average degree. The MaxInDeg($k$) family constructs the DAG sequentially: for each vertex $b$, it draws the number of parents from $\{0,\ldots,\min(k,b-1)\}$ and chooses that many parents among earlier vertices. The GridTree($k$) family starts from a directed $k \times k$ grid and repeatedly attaches new directed $k \times k$ grids along randomly selected sides, preserving the grid orientation, until the graph contains at least $n$ vertices. The Bipartite($p$) family splits the vertices into two layers and independently adds each left-to-right arc with probability $p$. The number of players ranges over $n \in \{5, 10, \ldots, 25\}$. To test the effect of soft priorities, we include the uniform-weight baseline and non-uniform settings where priority weights are drawn independently as $\lambda_i \sim \text{Unif}(1, R)$ with $R \in \{4, 16, 64, 256, 1024\}$. For each setting, $P^t_{\pi_0}$ is estimated using 10,000 independent chains, while the stationary pairwise probabilities $P^\star(i \prec_\pi j)$ are computed by exact dynamic programming, adapting the counting approach of Kangas et al. (2016) to the non-uniform target distribution. Concretely, for a feasible subset $S$, let $M(S) = \max_\preceq(S)$ and define the backward-removal weight

$$w(x; S) = \frac{\lambda_x|M(S)|}{\sum_{j\in M(S)}\lambda_j}, \qquad x \in M(S).$$

The unweighted extension count is then replaced by a weighted mass $Z(S)$ satisfying

$$Z(S) = \sum_{x\in M(S)} w(x; S)Z(S \setminus \{x\}), \qquad Z(\emptyset) = 1, \tag{29}$$

---

**Algorithm 2** Exact pairwise probability under the PASV target

---

**Input:** poset $([n], \preceq)$; priority weights $\lambda$; ordered pair $(a, b)$.
**Output:** $P^\star(a \prec_\pi b)$ under $p^{(\preceq, \lambda)}$.
Compute the transitive closure of $\preceq$.
If $a \preceq b$, **return** 1; if $b \preceq a$, **return** 0.
Save $Z(S)$ over feasible subsets using (29).
Save $F_\eta(S)$ using (30).
**return** the value in (31).

---

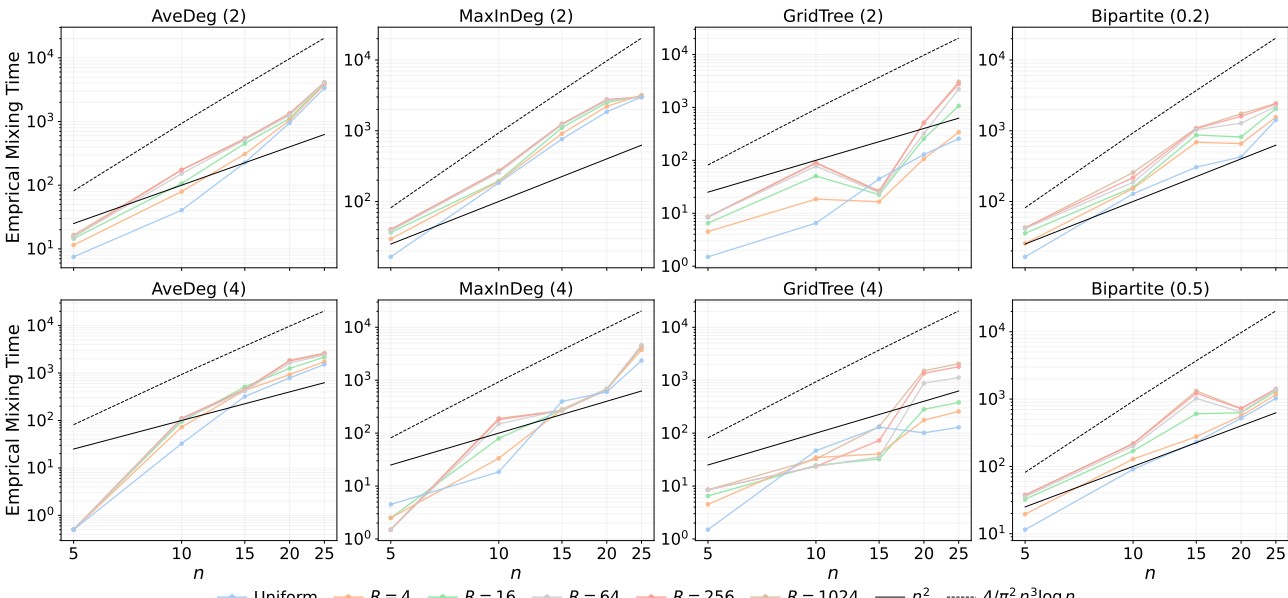

*Figure 8.* Empirical mixing time of the adjacent-swap M-H chain. Four synthetic poset families from Talvitie et al. (2017) are considered, each with two sub-settings: AveDeg, MaxInDeg, GridTree, and Bipartite. Both axes are log-scaled, so the slope indicates the exponent $c$ in an empirical $O(n^c)$ rate. Priority weights include the uniform baseline and $\lambda_i \overset{\text{iid}}{\sim} \text{Unif}(1, R)$ with $R \in \{4, 16, 64, 256, 1024\}$. The solid black line shows $n^2$, and the dashed black line shows the theoretical worst-case reference $\frac{4}{\pi^2} n^3 \log n$.

For an ordered pair $(a, b)$, the same backward-removal recursion is augmented with event states. Let $F_\eta(S)$ denote the unnormalized mass for the event $a \prec_\pi b$ when $\eta = 0$ means undecided, $\eta = 1$ means already true, and $\eta = 2$ means already false. For incomparable $a$ and $b$, the recursion is

$$F_1(S) = Z(S), \qquad F_2(S) = 0,$$

$$F_0(S) = \sum_{x \in M(S)} w(x; S) \begin{cases} F_1(S \setminus \{x\}), & x = b, \\ F_2(S \setminus \{x\}), & x = a, \\ F_0(S \setminus \{x\}), & x \notin \{a, b\}. \end{cases} \tag{30}$$

The desired stationary pairwise probability is then

$$P^\star(a \prec_\pi b) = \frac{F_0([n])}{Z([n])}. \tag{31}$$

Algorithm 2 summarizes the resulting computation for one ordered pair; applying it to all unordered pairs gives the matrix of stationary pairwise probabilities used in (28).

Figure 8 shows that the empirical mixing time is much closer to the $O(n^2)$ behavior reported by Talvitie et al. (2017) than to the worst-case $O(n^3 \log n)$ reference line. The results also indicate that the non-uniformity induced by $\lambda$ has limited effect on the mixing time in these settings. Thus, for the regimes considered here, the MCMC burn-in is not the dominant computational cost; the main cost comes from evaluating $U(S)$ along retained permutations.

### D.1.2. APPROXIMATION ERROR OF PASV

We next study the accuracy of the Monte Carlo estimator after permutations have been sampled. Unlike the application experiments, this simulation uses utility functions for which the true PASV can be computed in closed form. This allows us to measure estimation error directly rather than relying on qualitative agreement with domain expectations.

The utility is a sum-of-unanimity (SOU) game,

$$U(S) = \sum_{j=1}^{d} \alpha_j \mathbb{1}\{T_j \subseteq S\}, \qquad T_j \subseteq [n], \tag{32}$$

where $\alpha_j \overset{\text{iid}}{\sim} \text{Unif}(0.5, 1.5)$. SOU games provide a common synthetic benchmark in cooperative game experiments because they combine elementary unanimity components while preserving analytically tractable values; recent Shapley-value benchmarks also use this class of games (Li & Yu, 2024; Lee et al., 2025). We consider two synthetic scenarios. In Scenario 1, the player set is partitioned into $K$ equal-sized blocks $\{B_k\}_{k=1}^{K}$. Treating blocks as nodes, we add an edge $B_k \to B_\ell$ with probability 0.8 for $k < \ell$, and then set $i \preceq j$ for players $i \in B_k$, $j \in B_\ell$ whenever this block edge is present. Here $\lambda \equiv 1$, $d = K$, and $T_j = B_j$, so the true value is $\psi_i(U) = \alpha_j / |T_j|$ for $i \in T_j$. In Scenario 2, the DAG is the ordered partition $B_1 \prec \cdots \prec B_K$. We consider three weight settings, $\lambda \equiv 1$, $\lambda_i \sim \text{Unif}(1, 10)$, and $\lambda_i \sim \text{Unif}(1, 100)$. In this scenario, $d = n^2$ and each $T_j$ is sampled uniformly from $2^{[n]}$. For any nonempty $T$, define $r(T) = \max\{k : T \cap B_k \neq \emptyset\}$ and $M_T = T \cap B_{r(T)}$. The closed-form PASV contribution of the nonconstant SOU terms is then

$$\psi_i(U) = \sum_{j: i \in M_{T_j}} \alpha_j \frac{\lambda_i}{\sum_{r \in M_{T_j}} \lambda_r}. \tag{33}$$

Both scenarios use $n \in \{128, 512, 2048, 8192\}$, $K = n/16$, and equal-sized blocks. The maximum number of retained Monte Carlo samples is $N_{\text{MC}} = 10^4$, with burn-in 100,000 and thinning $10^3$. When the same subset $S$ is encountered repeatedly, we cache $U(S)$ and reuse the stored value. The four reported cases are Scenario 1 with $\lambda \equiv 1$, and Scenario 2 with $\lambda \equiv 1$, $\lambda_i \sim \text{Unif}(1, 10)$, and $\lambda_i \sim \text{Unif}(1, 100)$. For $m$ retained permutations, we report the absolute relative error

$$\text{ARE}(m) = \frac{\|\widehat{\psi}^{(m)} - \psi^\star\|_2}{\|\psi^\star\|_2}, \tag{34}$$

where $\psi^\star$ is the closed-form PASV and $\widehat{\psi}^{(m)}$ is the estimate based on $m$ retained permutations. We also report the area under the convergence curve,

$$\text{AUCC} = \frac{1}{100} \sum_{\ell=1}^{100} \text{ARE}(100\ell), \tag{35}$$

the number of unique utility evaluations, the runtime spent on utility evaluations, and the runtime spent on all other operations.

Figure 9 shows that the basic estimator approximates the true PASV well with a few hundred retained permutations across the simulated settings. The experiment also illustrates how the computational burden should be interpreted in applications. The Monte Carlo stage nominally traverses $O(N_{\text{MC}}n)$ prefix subsets, but repeated subsets can be cached, so the number of unique utility calls can be substantially smaller. When a downstream utility call is expensive, the total runtime can therefore be decomposed into the number of unique utility evaluations times the cost per evaluation, plus the sampler and bookkeeping overhead shown separately in the figure. The results indicate that the estimator remains practical for player sets with several thousand players in these controlled benchmarks.

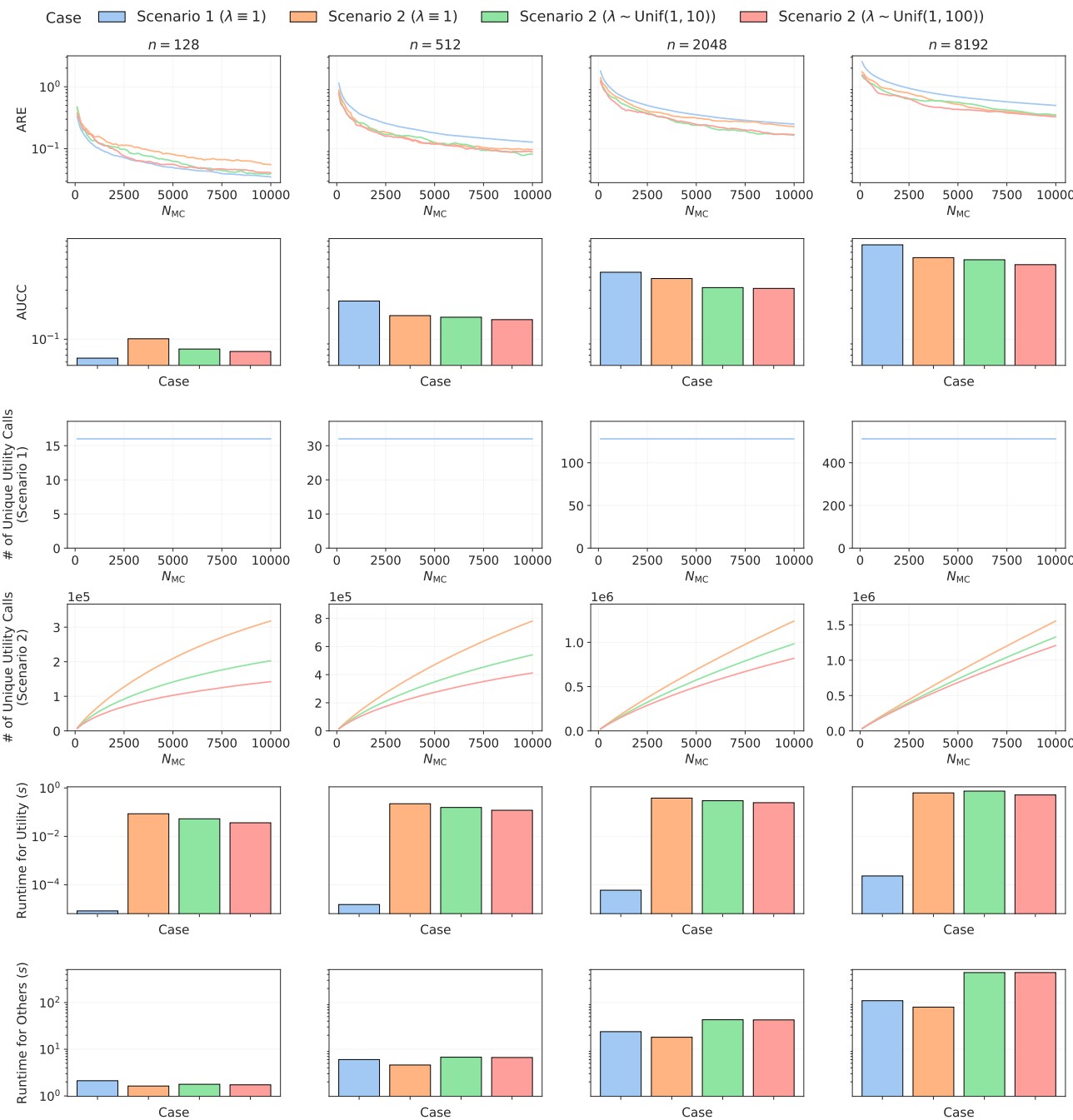

*Figure 9.* Approximation quality and computational cost of the basic PASV Monte Carlo estimator. The figure summarizes results across $n$, $N_{\mathrm{MC}}$, and the four cases described in Appendix D.1.2. The top two rows report ARE and AUCC, the middle rows report the number of unique utility evaluations, and the bottom rows separate utility-evaluation runtime from all other runtime. Utility values are cached and reused when the same subset is revisited.

## D.2. Proxy-Based Estimation Method

The previous subsection studies the direct Monte Carlo estimator, which estimates PASV by repeatedly evaluating marginal contributions along sampled linear extensions. This approach is simple and broadly applicable, but its statistical efficiency can be limited when the utility function has systematic low-order structure that is not explicitly used. The proxy-based estimator studied here follows the proxy-plus-residual strategy of regression MSR (Witter et al., 2026): fit a tractable surrogate for the utility, compute or estimate the PASV of that surrogate, and use Monte Carlo only to correct the residual. Its surrogate model is related to regression-based Shapley estimators: a linear proxy corresponds to the KernelSHAP viewpoint (Lundberg & Lee, 2017; Covert & Lee, 2021), while including higher-order interaction terms follows the polynomial extension in PolySHAP (Fumagalli et al., 2026). We adapt these ideas to precedence-constrained PASV by using a quadratic proxy over incomparable pairs and by reusing each evaluated utility value when estimating multiple player-value coordinates, in the spirit of maximum sample reuse (Wang & Jia, 2023). The goal of this experiment is not to replace the direct estimator in all regimes, but to understand when a fitted proxy can reduce approximation error under a fixed computational budget.

The proxy takes the form

$$\widehat{U}(S) \approx U(\emptyset) + \sum_{i \in S} a_i + \sum_{\{i,j\} \subseteq S,\ i \| j} b_{ij}, \tag{36}$$

where $i \| j$ denotes that neither $i \preceq j$ nor $j \preceq i$, and $b_{ij} = b_{ji}$. For pairs that are comparable in the DAG, the quadratic coefficient is set to zero. This choice keeps the proxy aligned with the precedence structure, since only incomparable pairs can vary in their relative sampled order. For this quadratic proxy, the PASV contribution has the closed form

$$\psi_i(\widehat{U}) = a_i + \sum_{j:j \neq i,\ i \| j} q_{ji} b_{ij}, \qquad q_{ji} := \mathbb{P}_{\pi \sim p^{(\preceq,\lambda)}}(j \prec_\pi i). \tag{37}$$

The estimator proceeds in three parts. First, the MCMC sampler is run to stationarity as in Algorithm 1. Second, sampled permutations and their prefix sets are used to evaluate $U$, fit $\widehat{U}$ by weighted least squares, and estimate the pairwise probabilities $q_{ji}$. Third, using linearity,

$$\psi(U) = \psi(\widehat{U}) + \psi(U - \widehat{U}),$$

we estimate the residual term $\psi(U - \widehat{U})$ with the direct Monte Carlo estimator.

We evaluate this proxy-based estimator on Scenario 2 from Appendix D.1.2 with $\lambda \equiv 1$. The player sizes are $n \in \{256, 512, 1024, 2048\}$. The proxy-fitting budget is $N_{\text{coef}} = kn$ subsets with $k \in \{0, 1, 2, 5\}$, where $k = 0$ corresponds to the direct estimator without a proxy. The probability-estimation budget is $N_{\text{prob}} = 100n$, and the residual-correction Monte Carlo budget is $N_{\text{MC}} = 10{,}000 - N_{\text{coef}}$. We also vary $\rho \in \{0, 0.5, 1\}$, which controls the proportion of discovered quadratic terms included in the proxy. The corresponding computational costs decompose as follows. The MCMC component costs $O(Nn)$ for $N$ iterations or samples. Utility evaluation costs $O(N_{\text{coef}}n)$ for proxy fitting plus $O((N - N_{\text{coef}})n)$ for residual correction, hence $O(Nn)$ in total. The additional weighted least-squares proxy fit costs $O(N_{\text{coef}}d^2)$, where $d$ is the number of proxy coefficients and can be as large as $O(n^2)$.

Figure 10 shows that the proxy can improve approximation accuracy when the proxy-fitting cost is not too large relative to the total budget. For $n \leq 512$, the settings $\rho \in \{0.5, 1\}$ consistently improve over the direct estimator under the same budget. The setting $\rho = 0$, which removes the quadratic terms, performs worst in this benchmark, indicating that pairwise terms are important for this non-uniform distribution over admissible orders. The case $\rho = 0.5$ often matches the accuracy of $\rho = 1$ while reducing overhead, making it a practical compromise in these simulations. For $n \geq 1024$, the proxy budget becomes more consequential. When $k \leq 2$, the proxy-based estimator still matches or improves on the direct estimator, whereas $k = 5$ can allocate too much of the budget to proxy fitting and make the direct estimator preferable.

Figures 11 and 12 show the corresponding computational tradeoff. The extra proxy runtime grows with $n$ and $k$, and $\rho = 0.5$ has substantially smaller fitting cost than $\rho = 1$. Peak memory also increases with $n$, with the largest $(n, \rho) = (2048, 1)$ cases requiring tens of GB during proxy construction. After the proxy has been fitted, the memory cost drops sharply and is much less sensitive to $\rho$. Overall, these results suggest that the proxy-based estimator is useful when moderate quadratic structure captures much of the utility variation, but its fitting budget and peak memory should be controlled for larger $n$.

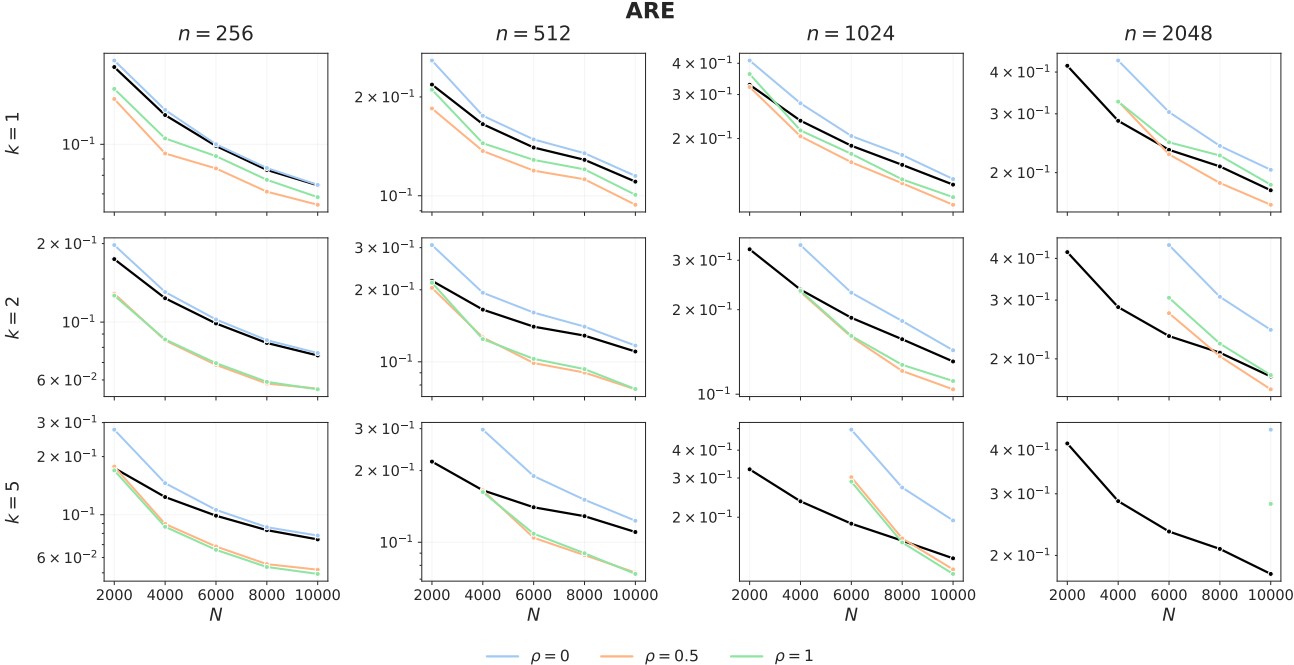

*Figure 10.* Approximation error of the proxy-based estimator. ARE is plotted as the Monte Carlo budget increases. The black curve is the direct estimator, while colored curves use the quadratic regression proxy with different values of $\rho$. Rows correspond to $k$, where $N_{\mathrm{coef}} = kn$; points are absent when the available budget is smaller than the required proxy-fitting budget.

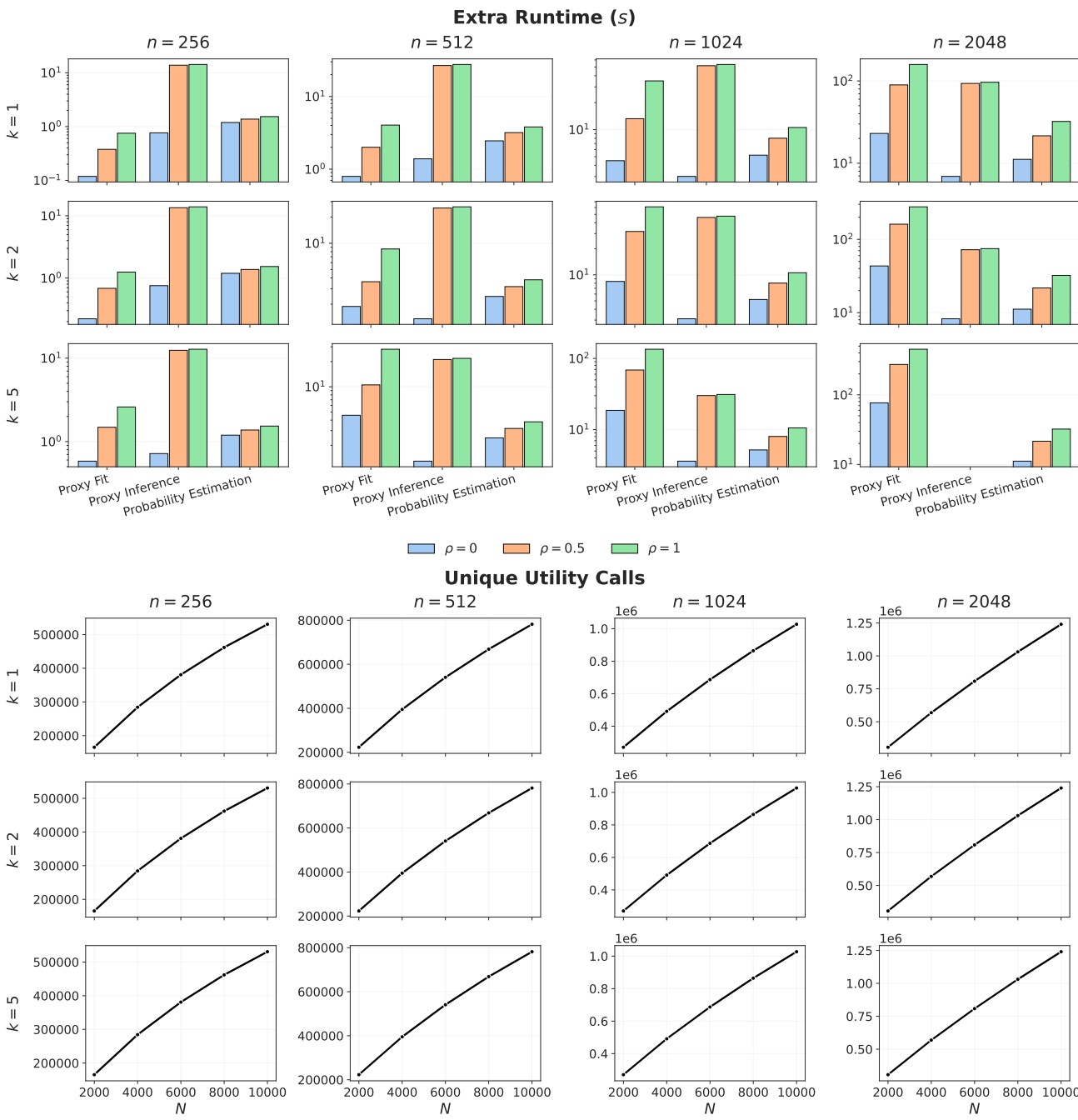

*Figure 11.* Extra runtime and unique utility evaluations of the proxy-based estimator. The top panel decomposes extra runtime into proxy fitting, proxy inference, and probability estimation. The bottom panel reports the number of unique utility evaluations. The unique utility evaluation curves coincide across $k$ and $\rho$, because this quantity is driven by $n$ and the fixed total budget rather than the proxy specification.

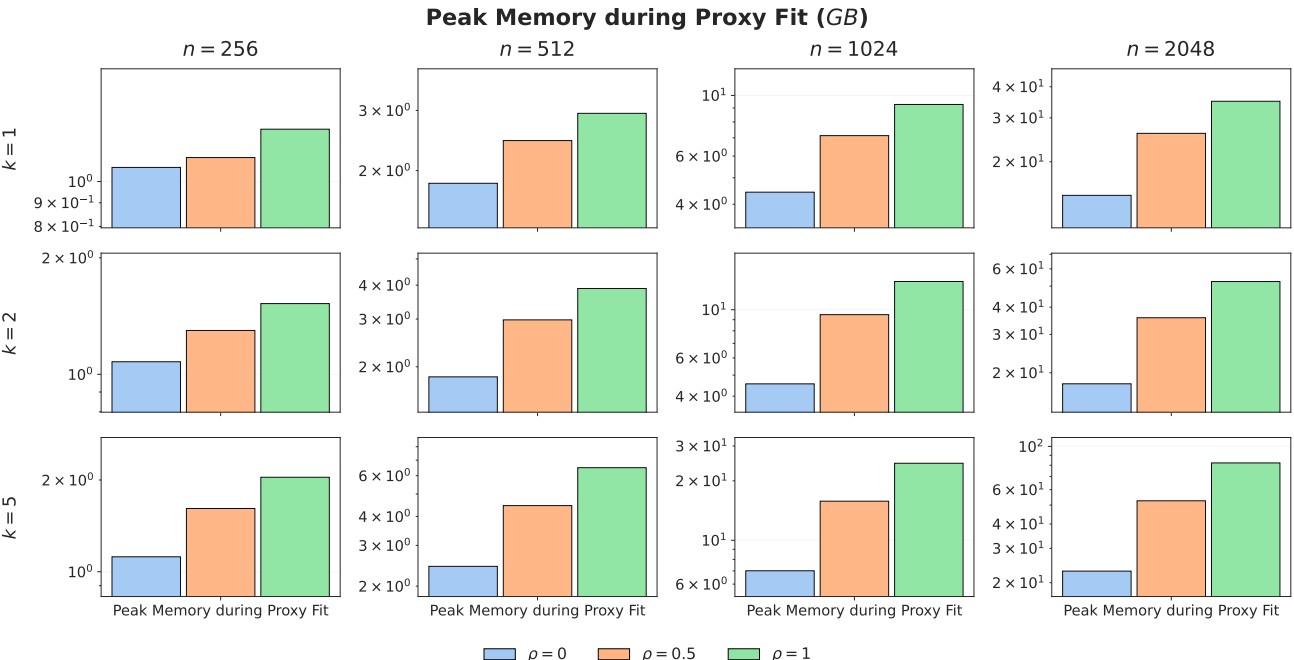

*Figure 12.* Peak memory during proxy fitting. The plot uses the same grid layout as Figure 10 and reports peak memory in GB during the proxy-fitting step for $\rho \in \{0, 0.5, 1\}$.

# E. Experimental Details in Section 5.1

All experiments were run at the Unity high-performance computing cluster, provided by the College of Arts and Sciences at The Ohio State University. Generative AI training and sampling in Section 5.1 used one NVIDIA V100 GPU with 32 GB VRAM per run. All other experiments were performed on CPU nodes with allocated resources fixed at 16 CPU cores and 32 GB RAM per run, using various Intel Xeon processors.[2]

## E.1. Detailed Setup

We provide additional implementation details for the data valuation experiments in Section 5.1. The random seed was fixed for dataset construction, and value estimation was replicated across 10 independent random seeds. We report the mean and one standard deviation across these runs.

Utility was defined as the test accuracy of a $k$-NN classifier with Euclidean distance using $k = 20$. For CIFAR10, we used ImageNet-pretrained ResNet-18 features, extracting the 512-dimensional representation from the global average pooling output immediately before the final fully connected layer. Each CIFAR10 image was normalized using ImageNet statistics prior to feature extraction, and the resulting embeddings were $\ell_2$-normalized.

Provider datasets were constructed as follows. The **Owner** dataset was sampled from the original dataset with balanced class proportions. The **Anchor** dataset was obtained by applying geometric and optical augmentations to the Owner samples. Each augmentation parameter was sampled independently from a uniform distribution over the stated range.

- Scaling in $[0.95, 1.05]$

- Brightness in $[0.7, 1.3]$ for MNIST, and in $[0.8, 1.2]$ for CIFAR10

- Contrast in $[0.7, 1.3]$ for MNIST, and in $[0.8, 1.2]$ for CIFAR10

- Translation sampled independently in the horizontal and vertical directions, each by up to $1/32$ of the image size.

- Rotation in $[-15°, 15°]$ for MNIST

- For MNIST, Gaussian blur with kernel size 3 and $\sigma \in [0.5, 1.5]$.

- For CIFAR10, horizontal flip with probability $0.5$.

For each **Booster**, we trained a class-conditional generative model on the Owner dataset, and also trained a second model on the Anchor-augmented version of the Owner dataset. We then sampled a class-balanced set of size $n/2$ from each model via label conditioning. Full generative model specifications are provided in Appendix E.2. The **Copier** dataset was formed by copying half of the Owner samples while preserving class balance, and likewise half of the Anchor samples. The **Poisoner** dataset was formed by taking half of the Owner samples with balanced class proportions and replacing each label with a uniformly sampled incorrect class label, and likewise for half of the Anchor samples.

All values were estimated via Monte Carlo using $N_{\mathrm{MC}} = 10{,}000$ samples. For PASV, we sampled linear extensions using Algorithm 1 with burn-in $B = 100{,}000$ and thinning $\tau = 10{,}000$. The index proposal sampler $f$ was uniform over $[n-1]$.

## E.2. Model Configurations for Generative AIs

For each generative model, we trained two separate class-conditional models, one using the original dataset and another using the Anchor-augmented version of the dataset. The resulting Booster dataset was formed by drawing $n/2$ samples from each of the two models, sampling uniformly across classes via label conditioning. All models were trained with Adam, using $(\beta_1, \beta_2) = (0, 0.9)$ for GAN and $(\beta_1, \beta_2) = (0.9, 0.999)$ otherwise. The following summarizes the model configurations used for each model.

For **GAN**, we used a DCGAN-style architecture based on transposed convolutions. We trained the model using hinge loss. We used learning rates $10^{-4}$ for the generator and $2 \cdot 10^{-4}$ for the discriminator. We used batch size 128 and trained for 40 epochs on MNIST and 60 epochs on CIFAR10. The latent dimension was set to 128 for MNIST and 256 for CIFAR10.

---

[2]Jobs could be placed on different nodes depending on cluster availability, so the specific CPU model varied across runs.

For **DDPM**, we used a class-conditional UNet with base width 64 on MNIST and 128 on CIFAR10. We used a linear noise schedule with $\beta_{\text{start}} = 10^{-4}$ and $\beta_{\text{end}} = 0.02$, with $T = 1000$ diffusion steps. Training used learning rate $10^{-4}$ and batch size 64, for 30 epochs on MNIST and 200 epochs on CIFAR10. Sampling used the full DDPM sampler with 1000 steps.

For **DDIM**, we sampled from the same trained DDPM parameters, conditioning on class labels in the same manner as DDPM. We used 250 sampling steps with a linear index schedule on MNIST and an index schedule uniform in the $\bar{\alpha}$ space on CIFAR10.

For **FM**, we used the same class-conditional UNet backbone as in DDPM. Training used learning rate $10^{-4}$ and batch size 64, for 30 epochs on MNIST and 200 epochs on CIFAR10, using time scale 1000. Sampling solved the associated ODE using the Euler method with 50 steps on MNIST and 200 steps on CIFAR10.

We provide sample visualizations for MNIST and CIFAR10 in Figures 13 and 14, respectively.

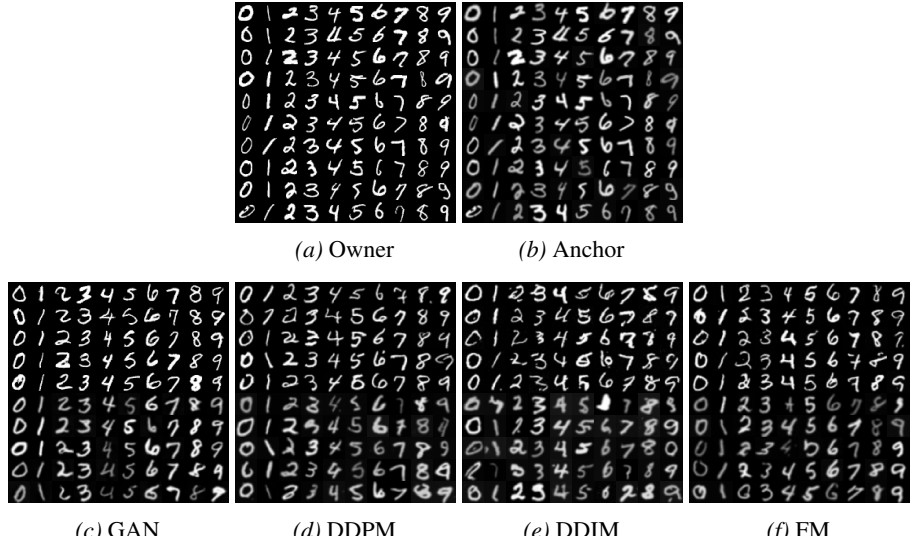

*(a)* Owner      *(b)* Anchor

*(c)* GAN      *(d)* DDPM      *(e)* DDIM      *(f)* FM

*Figure 13.* MNIST samples. For each generative model panel (c) to (f), the top five rows show samples generated from the model trained on the original dataset, and the bottom five rows show samples generated from the model trained on the Anchor augmented dataset.

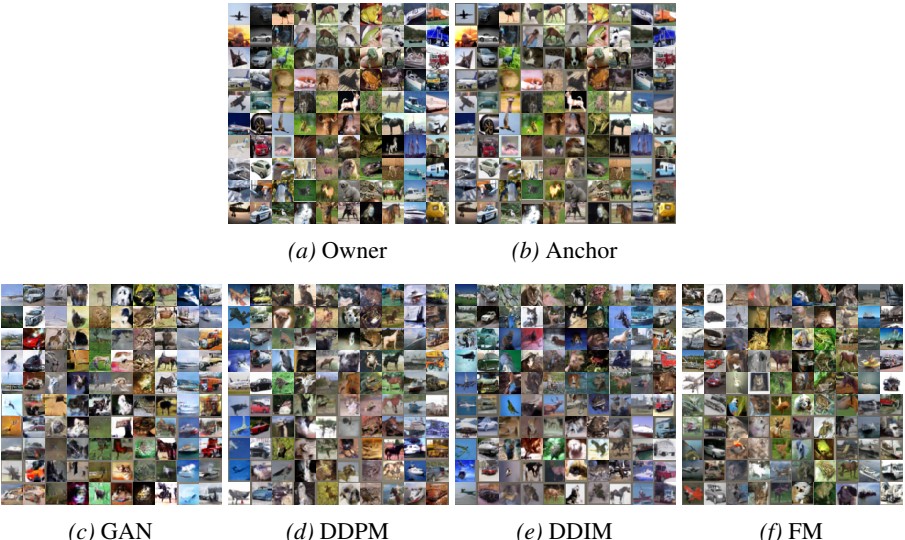

*(a)* Owner      *(b)* Anchor

*(c)* GAN      *(d)* DDPM      *(e)* DDIM      *(f)* FM

*Figure 14.* CIFAR10 samples. For each generative model panel (c) to (f), the top five rows show samples generated from the model trained on the original dataset, and the bottom five rows show samples generated from the model trained on the Anchor augmented dataset.

## E.3. Additional Results

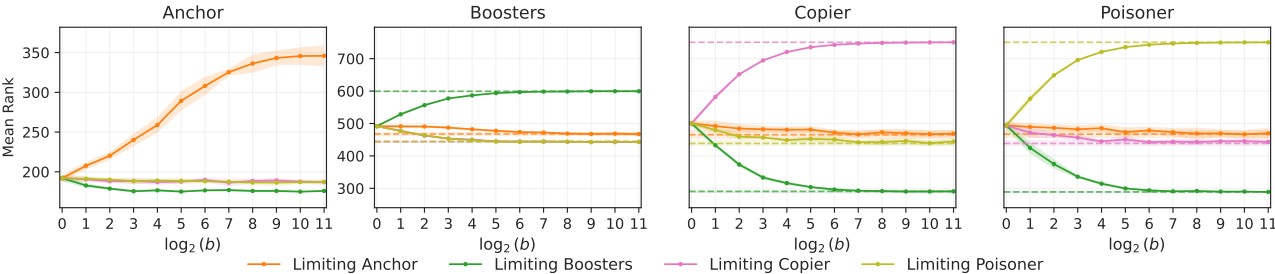

*Figure 15.* Change of mean rank by limiting weight of each provider.

*Table 1.* Provider-wise values on MNIST (mean (sd)).

| Provider | SV | WSV | PSV | PASV ($b=2$) | PASV ($b=8$) | PASV ($b=32$) |
|---|---|---|---|---|---|---|
| Owner | 0.0957 (0.0002) | 0.5270 (0.0000) | 0.5270 (0.0000) | 0.5270 (0.0000) | 0.5270 (0.0000) | 0.5270 (0.0000) |
| Anchor | 0.1148 (0.0004) | 0.0468 (0.0002) | 0.0927 (0.0056) | 0.0826 (0.0069) | 0.0673 (0.0040) | 0.0548 (0.0049) |
| GAN | 0.1225 (0.0003) | 0.0596 (0.0001) | 0.0483 (0.0028) | 0.0513 (0.0027) | 0.0512 (0.0021) | 0.0562 (0.0021) |
| DDPM | 0.1144 (0.0003) | 0.0507 (0.0002) | 0.0470 (0.0036) | 0.0557 (0.0039) | 0.0591 (0.0061) | 0.0624 (0.0047) |
| DDIM | 0.1078 (0.0003) | 0.0518 (0.0002) | 0.0330 (0.0034) | 0.0283 (0.0045) | 0.0281 (0.0079) | 0.0263 (0.0032) |
| FM | 0.1506 (0.0004) | 0.0825 (0.0003) | 0.0773 (0.0048) | 0.0767 (0.0035) | 0.0884 (0.0054) | 0.0960 (0.0070) |
| Copier | 0.0848 (0.0003) | 0.0073 (0.0002) | -0.0032 (0.0022) | -0.0065 (0.0012) | -0.0054 (0.0005) | -0.0061 (0.0006) |
| Poisoner | 0.0175 (0.0003) | -0.0178 (0.0002) | -0.0141 (0.0052) | -0.0072 (0.0015) | -0.0077 (0.0009) | -0.0086 (0.0012) |

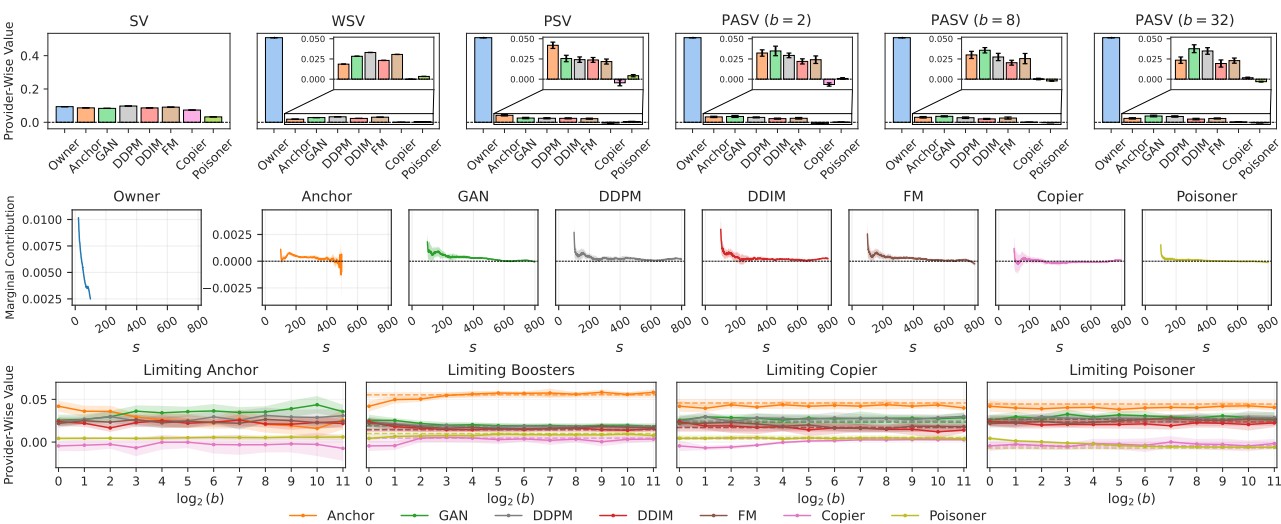

*Figure 16.* CIFAR10 results. **(Top)** Provider-wise values. **(Middle)** Marginal contribution versus subset size. **(Bottom)** Sensitivity analysis.

*Table 2.* Provider-wise values on CIFAR10 (mean (sd)).

| Provider | SV | WSV | PSV | PASV ($b$=2) | PASV ($b$=8) | PASV ($b$=32) |
|---|---|---|---|---|---|---|
| Owner | 0.0939 (0.0003) | 0.5080 (0.0000) | 0.5080 (0.0000) | 0.5080 (0.0000) | 0.5080 (0.0000) | 0.5080 (0.0000) |
| Anchor | 0.0866 (0.0003) | 0.0186 (0.0001) | 0.0419 (0.0039) | 0.0324 (0.0040) | 0.0300 (0.0045) | 0.0235 (0.0040) |
| GAN | 0.0835 (0.0002) | 0.0283 (0.0001) | 0.0256 (0.0040) | 0.0349 (0.0059) | 0.0358 (0.0033) | 0.0377 (0.0049) |
| DDPM | 0.0977 (0.0004) | 0.0330 (0.0001) | 0.0242 (0.0034) | 0.0294 (0.0029) | 0.0274 (0.0045) | 0.0349 (0.0041) |
| DDIM | 0.0864 (0.0003) | 0.0232 (0.0001) | 0.0238 (0.0030) | 0.0220 (0.0032) | 0.0204 (0.0030) | 0.0194 (0.0044) |
| FM | 0.0910 (0.0003) | 0.0306 (0.0001) | 0.0218 (0.0031) | 0.0241 (0.0047) | 0.0255 (0.0063) | 0.0229 (0.0032) |
| Copier | 0.0736 (0.0004) | -0.0002 (0.0001) | -0.0045 (0.0037) | -0.0066 (0.0020) | 0.0002 (0.0009) | 0.0017 (0.0007) |
| Poisoner | 0.0324 (0.0002) | 0.0034 (0.0001) | 0.0043 (0.0014) | 0.0009 (0.0010) | -0.0022 (0.0004) | -0.0031 (0.0005) |

---

**Algorithm 3** Utility evaluation with conditional $k$NN imputation.

---

**Input:** training set $\mathcal{D}_{\text{train}} = \{(x'_i, y'_i)\}_{i=1}^{n_{\text{train}}}$; test set $\mathcal{D}_{\text{test}} = \{(x_i, y_i)\}_{i=1}^{n_{\text{test}}}$; trained predictor $f$; subset $S \subseteq [n]$; number of neighbors $k$; number of evaluation points $n_{\text{eval}}$.

**Initialize** $\widehat{U}(S) \leftarrow 0$

**for** $t = 1$ **to** $n_{\text{eval}}$ **do**

    Sample $(x, y)$ uniformly from $\mathcal{D}_{\text{test}}$

    Find $k$ nearest neighbors $\{x^{(1)}, \ldots, x^{(k)}\}$ of $x$ among $\{x'_i\}_{i=1}^{n_{\text{train}}}$ restricted to coordinates in $S$

    **Initialize** $\widehat{u} \leftarrow 0$

    **for** $j = 1$ **to** $k$ **do**

        Form composite input $\widetilde{x}^{(j)} \leftarrow x_S \oplus x_{S^c}^{(j)}$

        $\widehat{u} \leftarrow \widehat{u} + f_y(\widetilde{x}^{(j)})$

    **end for**

    $\widehat{U}(S) \leftarrow \widehat{U}(S) + \widehat{u}/k$

**end for**

$\widehat{U}(S) \leftarrow \widehat{U}(S)/n_{\text{eval}}$

**return** $\widehat{U}(S)$

---

# F. Experimental Details in Section 5.2

## F.1. On-Manifold Utility with Conditional $k$-NN Imputation

Feature attribution evaluates how a fixed trained predictor changes as features are revealed, so we define the utility of a feature subset through a masking-based prediction score. Let $f_y(x)$ denote the predicted probability assigned to class $y$ at input $x$.

For a subset of feature indices $S \subseteq [n]$ and a labeled test point $(x, y)$, the utility is computed by keeping the observed coordinates $x_S$ fixed and imputing the remaining coordinates $x_{S^c}$ from a conditional reference distribution rather than from the marginal (Janzing et al., 2020; Sundararajan & Najmi, 2020; Frye et al., 2021). This on-manifold construction is used to avoid implausible masked inputs that can arise under marginal imputation when features are dependent.

Concretely, we approximate the conditional distribution of the unobserved coordinates given the observed ones, $p(x_{S^c}|x_S)$, using $k$-NN sampling in the subspace restricted to coordinates in $S$. Given a test point $(x, y)$ and $S$, let $\{x^{(1)}, \ldots, x^{(k)}\}$ denote the $k$ nearest neighbors of $x$ among training inputs when distances are computed using only the coordinates in $S$. For each neighbor $x^{(j)}$, we form a composite input $x_S \oplus x_{S^c}^{(j)}$, which keeps the coordinates in $S$ from $x$ and fills the remaining coordinates from $x^{(j)}$. The per-point utility is defined as the conditional expectation

$$u_S(x, y) = \mathbb{E}_{x'_{S^c} \sim p(x'_{S^c}|x_S)}\big[f_y(x_S \oplus x'_{S^c})\big],$$

and is estimated using the $k$-NN conditional sampler by

$$\widehat{u}_S(x, y) = \frac{1}{k} \sum_{j=1}^{k} f_y(x_S \oplus x_{S^c}^{(j)}).$$

Finally, following the global aggregation used in Shapley-based feature attribution, we define the subset utility as

$$U(S) = \mathbb{E}_{(x,y) \sim p(x,y)}\big[u_S(x, y)\big].$$

In implementation, we estimate $U(S)$ by Monte Carlo using $n_{\text{eval}}$ test points $(x^{(t)}, y^{(t)})$ sampled uniformly from the test split and the $k$-NN estimator $\widehat{u}_S(x, y)$, i.e.,

$$\widehat{U}(S) = \frac{1}{n_{\text{eval}}} \sum_{t=1}^{n_{\text{eval}}} \widehat{u}_S\big(x^{(t)}, y^{(t)}\big).$$

Implementation details of the utility evaluation procedure are provided in Algorithm 3.

## F.2. Detailed Setup

We provide further implementation details for the feature attribution experiments in Section 5.2. We used a fixed random seed for the train test split, and replicated value estimation across 10 independent random seeds. In the results, we report the mean and one standard deviation across these runs.

The experimental setup largely followed Frye et al. (2020). The Census Income dataset was preprocessed by dropping samples with missing values, yielding 45,222 samples with 12 features (intentionally excluding `fnlwgt` and `education-num`), followed by a 75/25 train test split. All categorical features were encoded using one hot encoding, while continuous features were standardized to have zero mean and unit variance. An MLP classifier $f$ was trained with ReLU activations and $\ell_2$ regularization with coefficient $10^{-4}$. Optimization used Adam with $(\beta_1, \beta_2) = (0.9, 0.999)$ and learning rate $10^{-3}$. Training ran for up to 200 epochs with batch size 128, using early stopping with a validation fraction of 0.25 and a patience of 20 epochs. For the conditional sampler used to evaluate $U(S)$, we set $k = 100$, $n_{\text{eval}} = 1000$ and used Euclidean distance. All values were estimated based on $N_{\text{MC}} = 3000$ Monte-Carlo samples, and for the general DAG setting, linear extensions were sampled using Algorithm 1 with burn-in $B = 10,000$ and thinning $\tau = 1,000$.

## F.3. DAG Construction for Census Income Dataset via LLM Consultation

In this experiment, the detailed precedence DAG over the 12 Census Income features was obtained by starting from the coarse ordered-partition constraint of Frye et al. (2020) and consulting **ChatGPT 5.2 Thinking** only to propose candidate relations based on common sense and domain knowledge. Note that we are not making a causal claim providing this DAG, and the full conversation transcript is provided below for transparency.

---

**Prompt**

```
We consider the U.S. Census Income dataset for predicting whether annual income
exceeds $50K. Features included in the dataset are as below:

- age: age in years.
- workclass: employment type (e.g., Private, Self-emp, Gov).
- education: highest education level attained (categorical; e.g., Bachelors, HS-grad,
etc.).
- marital-status: marital status (e.g., Married-civ-spouse, Never-married, Divorced,
etc.).
- occupation: job category (e.g., Tech-support, Craft-repair, Exec-managerial, etc.).
- relationship: household relationship/role (e.g., Husband, Wife, Own-child, Not-in-
family, etc.).
- race: self-identified race category.
- sex: biological sex / gender as recorded (Male/Female).
- capital-gain: capital gains (numeric).
- capital-loss: capital losses (numeric).
- hours-per-week: usual hours worked per week (numeric).
- native-country: country of origin (categorical).

We treat each input variable as a player in a cooperative game and use a DAG over
features to encode precedence constraints for feature attribution. For constructing a
DAG, A common approach in existing papers is to partition variables into two blocks:
A (demographic): {age, race, sex, native-country}
B (socioeconomic): {workclass, education, marital-status, occupation, relationship,
capital-gain, capital-loss, hours-per-week}
and assume a complete bipartite DAG A -> B (i.e., every variable in A has a directed
edge into every variable in B), with no edges within A and no edges within B.

Please evaluate whether the DAG specified above is reasonable. If it is, simply say: "
The specified DAG is reasonable." and ignore everything below. If it is not, propose a
 more realistic DAG that better reflects plausible relationships among these variables.

If you propose a new DAG, first provide a brief paragraph explaining your design
principles for the DAG, and then provide the DAG as an edge list in the form (parent
-> child).
```

---

Response

The specified bipartite DAG A -> B is not very realistic. While it's sensible that
demographic variables often precede many socioeconomic outcomes, the assumption of no
edges within B is a major miss: there are strong, plausible precedence relations like
education -> occupation/workclass, marital-status -> relationship, and occupation/
workclass -> hours-per-week, etc. Treating all B variables as mutually "parallel" can
distort attributions by forcing effects that plausibly flow through education/
occupation to instead be credited directly to demographics.

Design principles

I'm aiming for a DAG that (1) respects temporal/structural precedence (stable
attributes and early-life factors precede later labor-market outcomes), (2) captures
the most obvious within-socioeconomic dependencies (especially education -> occupation
/workclass and marital-status -> relationship), (3) stays sparse and defensible (only
edges that are broadly plausible across the population), and (4) avoids dubious causal
 claims (e.g., race <-> native-country) by not forcing edges where directionality is
unclear.

Proposed DAG (edge list)

(age -> education)
(age -> marital-status)
(age -> occupation)
(age -> relationship)
(age -> capital-gain)
(age -> capital-loss)
(sex -> education)
(sex -> occupation)
(sex -> workclass)
(sex -> marital-status)
(sex -> relationship)
(race -> education)
(race -> occupation)
(race -> workclass)
(native-country -> education)
(native-country -> occupation)
(native-country -> workclass)
(education -> occupation)
(occupation -> workclass)
(occupation -> hours-per-week)
(workclass -> hours-per-week)
(education -> capital-gain)
(education -> capital-loss)
(occupation -> capital-gain)
(occupation -> capital-loss)
(marital-status -> relationship)

This keeps the intuitive "demographics often precede socioeconomic outcomes" idea, but
 also encodes key intra-B precedence constraints that the complete bipartite A -> B
structure cannot represent.

*Table 3.* Average marginal contribution of `capital-gain` under different ordering with `marital-status` and `capital-loss`; The baseline sampling distribution ($\lambda = 1$) is used. Entries report mean (sd) across replications.

| Case of Ordering | Proportion in Samples | Marginal Contribution |
|---|---|---|
| `marital-status` $\rightarrow$ `capital-gain` $\rightarrow$ `capital-loss` | 0.3873 (0.0087) | 0.01663 (0.00000) |
| $\{$`marital-status`, `capital-loss`$\} \rightarrow$ `capital-gain` | 0.4620 (0.0087) | 0.01807 (0.00001) |
| `capital-gain` $\rightarrow \{$`marital-status`, `capital-loss`$\}$ | 0.1134 (0.0062) | 0.01835 (0.00002) |
| `capital-loss` $\rightarrow$ `capital-gain` $\rightarrow$ `marital-status` | 0.0373 (0.0048) | 0.02040 (0.00004) |

## F.4. Additional Experiments

### F.4.1. SWEEP DIAGNOSTICS

As shown in Figure 17 (and in the right panel of Figure 7), we notice that varying the priority weights to promote or delay a given feature does not necessarily induce a monotone change in its estimated value. Motivated by these non-monotone responses, we conduct a diagnostic analysis and focus on a representative case that exhibits a obvious U-shaped pattern under the sweep, `capital-gain`. First, based on the general DAG in Figure 6b, we can anticipate what happens in each limiting case. Clearly, since `capital-gain` is maximal in the full player set, in the $\lambda \rightarrow \infty$ scenario it is placed at the very end of the linear extension with probability one. On the other hand, in the $\lambda \rightarrow 0^+$ case, `capital-gain` is promoted as much as possible until it becomes the unique element of the maximal set. In particular, in the perspective of the backward sampling scheme, $\{$`age`, `sex`, `native-country`, `race`, `education`, `occupation`$\}$ (and only these) always appear before it, so under the limiting distribution `capital-gain` occupies seventh position with probability one (which is also consistent with the rank pattern in Figure 18). Meanwhile, we observed the following interesting relationship among the three players `capital-gain`, `capital-loss`, and `marital-status`:

1. `capital-gain` attains a **larger** marginal contribution when `capital-loss` has already been included in the model.

2. `capital-gain` attains a **smaller** marginal contribution when `marital-status` has already been included in the model.

To support this, Table 3 partitions sampled linear extensions by the relative ordering among `capital-gain`, `capital-loss`, and `marital-status`, and reports the corresponding mean marginal contribution of `capital-gain`. The table shows a dependence among three features. When `marital-status` precedes `capital-gain` while `capital-loss` remains absent, the marginal contribution of `capital-gain` is the smallest, and this happens frequently. In contrast, when `capital-loss` precedes `capital-gain` and `marital-status` is still absent, `capital-gain` achieves the largest marginal contribution, although it occurs rarely. The mixed case, where `capital-gain` appears after both `marital-status` and `capital-loss` (as in $\lambda \rightarrow \infty$) or before both (as in $\lambda \rightarrow 0^+$) yield intermediate marginal contributions. Noting that the weight average of the marginal contribution in Table 3 is a value of `capital-gain` under $\lambda = 1$, this clarifies the U-shape behavior of the sweeping result. In summary, `capital-loss` acts as an amplifying context for `capital-gain`, whereas `marital-status` acts as a suppressing context, and the priority sweep modulates the value of `capital-gain` value primarily through shifting the frequencies of these contexts.

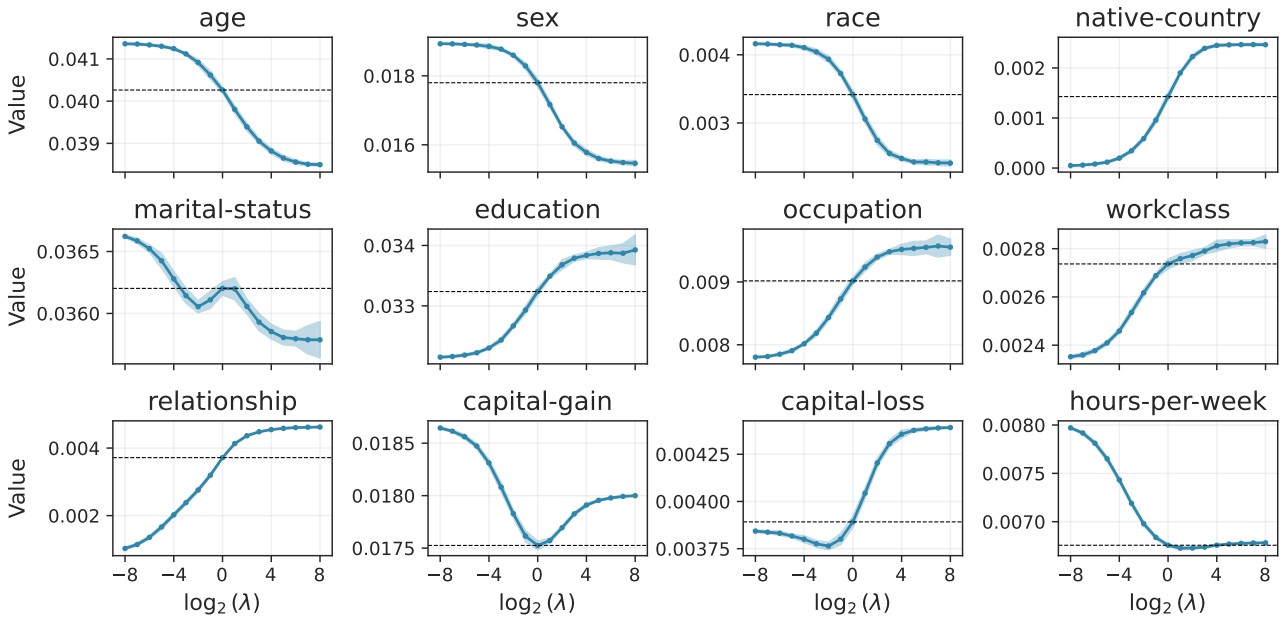

*Figure 17.* Change of feature value from priority sweep on the Census Income dataset.

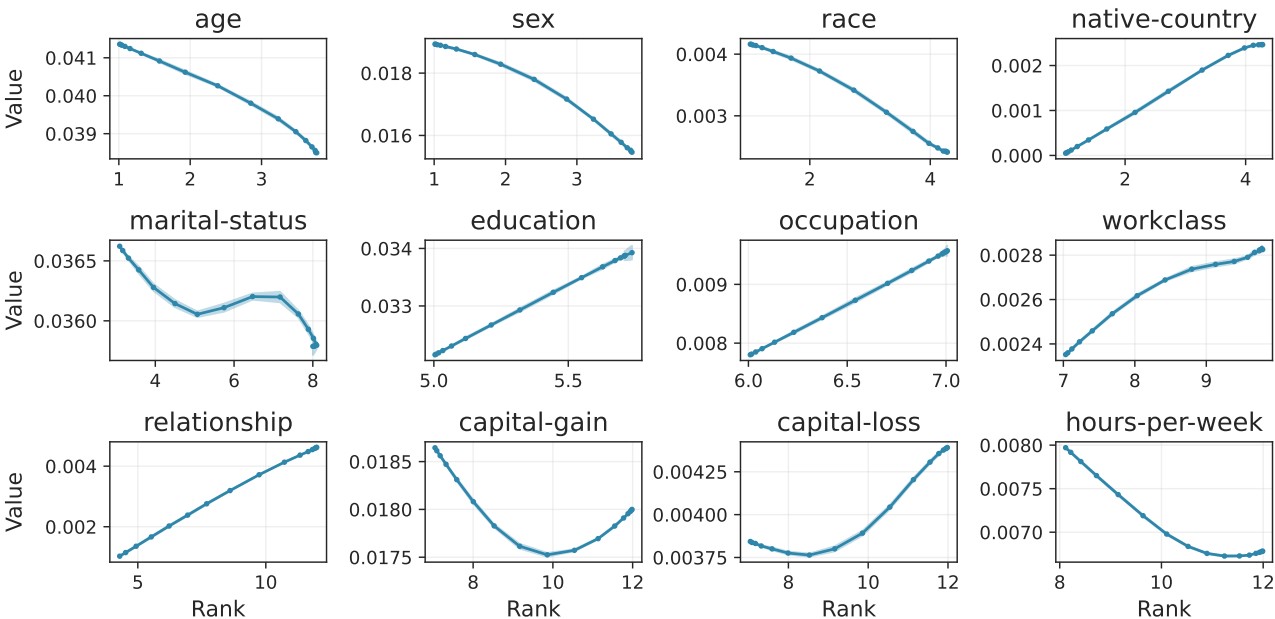

*Figure 18.* Feature values as a function of the feature's rank induced by priority sweep.

F.4.2. SENSITIVITY ANALYSIS

The feature-attribution experiment in Section 5.2 already shows that the attribution vector changes when the coarse ordered-partition DAG is replaced by the more detailed DAG in Figure 6b. Here we study a finer robustness question: how sensitive are the resulting PASV scores to local misspecification of the detailed DAG? This question is important because a precedence DAG is a modeling input rather than an observed ground truth. At the same time, complete insensitivity to the DAG would not be desirable: if a structural change substantially alters which feature can precede another, then a precedence-aware attribution method should reflect that semantic change. Thus, the purpose of this experiment is to distinguish small structural perturbations that leave the attribution stable from larger perturbations that materially change the induced order distribution.

We use the same Census Income feature-attribution setting as in Section 5.2. The base DAG is the general DAG in Figure 6b, which has 12 nodes and 26 edges, and we set $\lambda \equiv 1$. Only the DAG edges are perturbed; all other experimental configurations are kept fixed. Let $\psi$ and $\psi'$ denote the PASV vectors under the base and perturbed DAGs, respectively. For pairwise order probabilities, let

$$Q_{ij} = \mathbb{P}(i \prec_\pi j) \quad \text{and} \quad Q'_{ij} = \mathbb{P}'(i \prec_\pi j),$$

where the probabilities are estimated by Monte Carlo under the corresponding DAG.

We use Relative Attribution Shift (RAS) as the primary measure of attribution change:

$$\text{RAS} = \frac{\|\psi' - \psi\|_2}{\|\psi\|_2}. \tag{38}$$

To interpret RAS, we also report four auxiliary quantities: the Pearson correlation between $\psi$ and $\psi'$, the top-4 overlap between the four highest-valued features under the two vectors, the order-distribution change

$$\Delta_{\text{ord}} = \frac{1}{n(n-1)} \sum_{i \neq j} |Q'_{ij} - Q_{ij}|, \tag{39}$$

and the fraction of node pairs whose deterministic relation changes. For the latter, let $r_\preceq(i,j) \in \{i \preceq j,\ j \preceq i,\ i \parallel j\}$ denote the relation type of the unordered pair $\{i,j\}$ under the DAG. We report

$$\Delta_{\text{rel}} = \frac{2}{n(n-1)} \sum_{1 \leq i < j \leq n} \mathbb{1}\{r_{\preceq'}(i,j) \neq r_\preceq(i,j)\}. \tag{40}$$

We consider four perturbation scenarios. First, we delete a single effective edge, considering the 12 deletions corresponding to the transitive reduction. Second, we add a single edge, using 20 randomly sampled valid additions among currently incomparable pairs that preserve acyclicity. Third, we delete minimum edge cuts that disconnect the DAG into two equal components, yielding 6 cases. Fourth, we rank the 12 effective single-edge deletions by their RAS, and for $K \in \{3, 4, 5\}$ either delete the top-$K$ most influential edges or keep only those top-$K$ edges while deleting the others. The effective deleted edges and sampled valid additions used in the single-edge panels are listed below.

| Effective deletions | Effective deletions |
|---|---|
| marital-status→relationship | workclass→hours-per-week |
| education→occupation | sex→education |
| age→marital-status | occupation→capital-loss |
| sex→marital-status | native-country→education |
| age→education | occupation→capital-gain |
| occupation→workclass | race→education |

| Valid additions | Valid additions |
| --- | --- |
| relationship→education | relationship→capital-gain |
| relationship→occupation | capital-loss→capital-gain |
| marital-status→race | relationship→hours-per-week |
| marital-status→native-country | age→race |
| sex→age | marital-status→capital-gain |
| hours-per-week→marital-status | workclass→capital-loss |
| race→sex | workclass→capital-gain |
| relationship→workclass | capital-gain→hours-per-week |
| native-country→age | marital-status→hours-per-week |
| marital-status→occupation | race→relationship |

Figure 19 shows that PASV is stable to minor perturbations but sensitive to structural changes that substantially modify the admissible order distribution. Many single non-critical edge additions or deletions leave the attribution vector nearly unchanged, with RAS below 0.05 and Pearson correlation above 0.97. In contrast, disconnecting the DAG or removing critical edges produces larger shifts. The figure also shows that RAS is closely tied to changes in the pairwise order probabilities. This supports the intended interpretation of PASV under DAG uncertainty: robustness is expected when a perturbation has little effect on the order distribution, whereas a semantically consequential change in the precedence structure should be visible in the attribution.

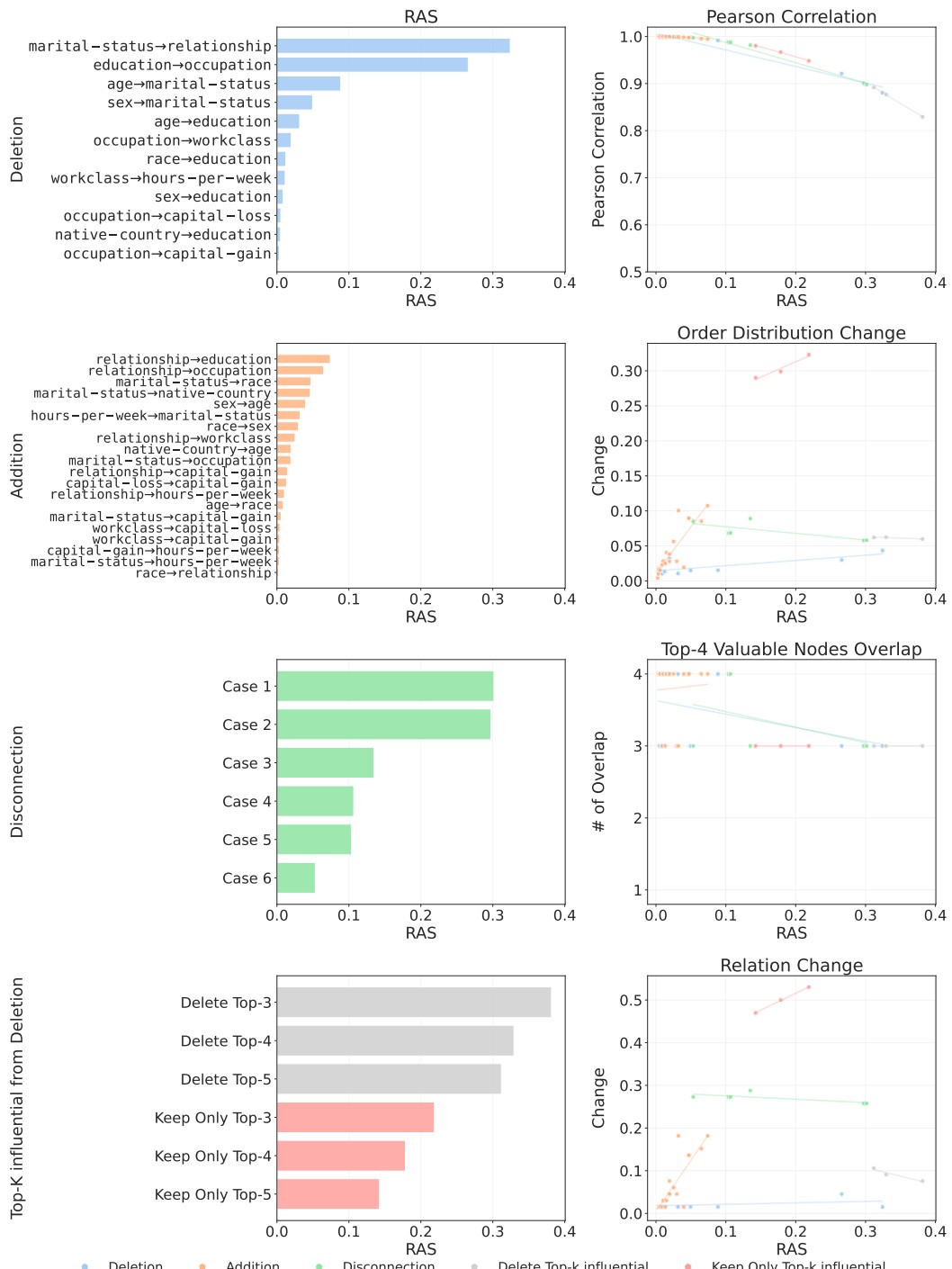

*Figure 19.* Sensitivity analysis under DAG misspecification in the Census Income feature-attribution experiment. The base DAG is Figure 6b with $\lambda \equiv 1$. Rows correspond to single-edge deletion, single-edge addition, DAG disconnection, and deleting or keeping only the top-$K$ influential edges. The left column reports RAS for each perturbation, and the remaining columns plot stability metrics against RAS.

