# OpenReview forum: "Priority-Aware Shapley Value"
_ICML.cc/2026/Conference — ICML 2026 regular_

### Official Review · Reviewer_uGr4 · 2026-03-04

**Soundness:** 4
**Presentation:** 4
**Significance:** 2
**Originality:** 3
**Overall Recommendation:** 5
**Confidence:** 4

**Summary:**

This paper proposes priority aware Shapley values (PASV) which alters the formulation and axiomatization of the Shapley value by encoding *precedence* of a player over other players (the tendency of a player to be added earlier/later into a coalition compared to other players). The paper formalizes PASV and conducts experiments in data valuation and feature attribution illustrating the usefulness of PASV as a diagnostics tool to analyze if precedence has a profound impact on the resulting SVs.

**Compliance With Llm Reviewing Policy:**

Affirmed.

**Final Justification:**

A technically solid paper which presents a novel and interesting new perspective into the Shapley value as an attribution method. It is not ground-breaking but nonetheless very interesting. A clear accept.

**Key Questions For Authors:**

Q1: What happens in the exponential complexity case of the players growing larger? Can you approximate PSAV efficiently, for example by adapting existing estimation methods for the Shapley Value [1-4]? I would like the authors to take this point into consideration.

**Limitations:**

The paper does not explicitly state its main limitation: The lack of an efficient estimator for its scores and thus being only applicable to low-player settings (10 features).

**Strengths And Weaknesses:**

**Summary**: I think this is a borderline paper on the side of accept with clear benefits, a proper organization/presentation, and good experimental evaluation. I only have one weakness: estimation of PASV, which would improve the paper and make it actually practically useful.

### Strengths
- Good Experiments. The experiments conducted in the paper show that PASV is useful as a diagnostics tool and that applications for the variation of the Shapley Value exists in different domains (data valuation and feature attribution) where PASV can help in a couple of interesting ways.
- Good Motivation: While not everywhere required, the paper motivates the need for considering precedence well.
- Strong theoretical justification. Adapting the definition of "worth" for the Shapley calculation is of course a very pervasive change targeting the core of the Shapley Value as a summary measure. However, the paper justifies these decisions formally and analyze its theoretical implications.
- Great presentation and writing. I want to explicitly point out how well the paper is written and presented. The paper is paced and flows very well and while it is very technical it is a pleasure to read!


### Weaknesses:
- **No experiments and results for larger number of players**: My only issue with the paper ist that all of the experiments and computations presented in this paper are only on very small-scale settings where the number of players (features / data vendors) is small such that computation of PSAV is still computationally feasible. This is of course quite a big limitation for a Shapley paper, which if addressed properly would improve the quality of the paper and its application substantially. However, I think this should be quite naturally be doable. I suppose transferring state-of-the-art methods such as the recently proposed Regression MSR [1] might be a bit hard. Also adapting KernelSHAP [2] could be problematic because of the side constraint ensuring efficiency. However, with permutation sampling [3] or the MSR principle [4] this should be achievable.
- Therein, the paper does not really include a discussion on the computational effort / complexity of computing its results. I think this is probably not an issue for small-scale settings, yet would still be interesting to note.

### References:
- 1: https://openreview.net/forum?id=Qabko39AS5
- 2: https://arxiv.org/abs/1705.07874
- 3: https://www.sciencedirect.com/science/article/abs/pii/S0305054808000804
- 4: https://arxiv.org/abs/2205.15466

---

> ### Author Rebuttal · Authors · 2026-03-31
>
> Thank you for the thoughtful review and suggestion for new methodology. Rebuttal figures:
>
> https://anonymous.4open.science/r/PASV-rebuttal/figures.pdf
>
> **Q1(a): "only applicable to low-player settings (10 features)"**
> We would like to gently note that our data valuation experiment (Section 5.1) already operates at n=800 players in a 14-block DAG; the 10-feature setting appears only in the feature attribution experiment (Section 5.2).
>
> We also add two new scalability experiments for:
> - **Our method 1** (original MCMC algorithm): see our reply to Reviewer aw3s.
> - **Our method 2** (adapting [1] via regression MSR): see below.
>
> **Q1(b): adapting existing estimation methods [1]–[4]**
>
> Thank you for this suggestion!
>
> - Following your advice, we adapted [1], using arxiv:2601.18608 (an extension of [2]);
> - Both our methods adopt the ideas of permutation sampling [3] and MSR [4];
> Our method 2 formulation:
> Proxy (quadratic regression MSR):
> $$
> \hat U(S) \approx U(\emptyset) + \sum_{i\in S} a_i + \sum_{\{i,j\}\subseteq S, i \\| j} b_{i,j},
> $$
> setting $b_{i,j}=b_{j,i}$, where $i \\| j$ denotes neither $i\prec j$ nor $j\prec i$.
> If a DAG edge exists between $(i,j)$, then we can set $b_{i,j}=b_{j,i}=0$
> PASV for proxy (closed-form):
> $$
> \psi_i(\hat U) = a_i + \sum_{j:j\neq i, i\\| j} \mathbb{P}(j\prec i\ \ \text{in }\pi) b_{i,j}.
> $$
> Bias correction:
> $$
> U(S) \approx \hat U(S) + \text{(bias correction)},
> $$
> where bias correction exactly follows [1], saving that we sample permutations, not sets.
>
> Algorithm sketch:
> - Part 0: run Stage 1 of our method 1 until MCMC reaches stationarity.
> - Part 1 (proxy): sample permutations from our MCMC sampler; use their prefix sets to evaluate $U$ and fit proxy $\hat U$ by weighted least square; estimate $\mathbb{P}(j\prec i\ \ \text{in }\pi)$ (feasible); plug in to get $\hat\psi(\hat U)$.
> - Part 2 (bias correction): by linearity, $\psi(U)=\psi(\hat U)+\psi(U-\hat U)$; estimate $\psi(U-\hat U)$ by our method 1 applied to $U-\hat U$.
>
> We add:
>
> **Added experiment RE4**
>
> for our method 2 (Figures R4-1, R4-2, R4-3):
>
> Setup: same synthetic benchmark as in our scalability experiment (see reply to Reviewer aw3s), Scenario 2, uniform $\lambda$. See figure captions for detailed settings.
>
> Control:
>
> - $n\in\\{256,..., 2048\\}$
> - $k\in\\{0,1,2,5\\}$ controls the proxy-fitting budget by setting $N_{coef}=kn$ subsets. ($k=0$ corresponds to our method 1)
> 	- This controls the number of rows of the design matrix for solving $(a,b)$.
> - $N_{prob}=100n$, $N_{MC}=10000-N_{coef}$
> - $\rho\in\\{0,0.5,1\\}$ controls how much proportion of the discovered quadratic terms would be included.
>
> Figure R4-1 (ARE):
> - Some **our method 2** curves start late. This is because the proxy fit demands $N\ge kn$ to collect the required number of permutations.
> - $n\leq 512$: **our method 2** with$\rho\in\\{0.5,1\\}$ consistently outperforms **our method 1** (black) under the same total budget, confirming the proxy effectively reduces ARE.
> - $\rho=0$ (KernelSHAP, blue) performs worst – quadratic terms are essential as the underlying distribution is not uniform on all $n!$ permutations.
> - $\rho=0.5$ often matches $\rho=1$ in ARE with less overhead, making it a practical default.
> - $n\geq 1024$: the proxy consumes a larger budget share. $k\leq 2$: **our method 2** still matches or improves on **our method 1**; $k=5$: proxy overhead dominates cost and **our method 1** can be preferable.
>
> Figure R4-2 (Extra runtime (s)):
> - Proxy overhead grows with $n,k$; $n\leq 512$: on the order of seconds.
> - $\rho=0.5$ roughly halves fitting cost vs. $\rho=1$.
> - The number of utility evaluations depends on $n$ but not $k$ and $\rho$.
>
> Figure R4-3 (Proxy memory (GB)):
> - Peak build-time memory grows with $n$; $(n,\rho)=(2048,1)$: tens of GB.
> - After fitting, memory cost drops sharply and is nearly independent of $\rho$.
> - Overall, $\rho=0.5$ substantially reduces peak memory while retaining most accuracy benefits.
>
> Computational complexity of each part of **our method 2**:
> - MCMC: each iteration costs $O(n)$; $N$ iterations/samples cost $O(Nn)$.
> - Utility evaluation: $O(N_{\rm coef}n)$ (part 1) + $(N-N_{\rm coef}) n$ (part 2) = $O(Nn)$ (total).
> - Proxy fit overhead: weighted least squares costs $O(N_{\rm coef} d^2)$, where $d=$~(number of proxy coefficients), up to $O(n^2)$.

---

> > ### Author Rebuttal · Reviewer_uGr4 · 2026-04-01
> >
> > Thank you! The larger player counts were not clear for me, which should have been made clearer in the submission I suppose. I very much like the responses regarding Regression MSR and the swift solutions you came up with. Good paper! I raised my score.

---

> > > ### Author Response · Authors · 2026-04-01
> > >
> > > Thank you very much!  We really appreciate your insightful comments that helped us improve the paper.
> > > Following your advice, in the next version of this paper, we will revise the player count statement in our Section 5.1 to make the message ``$n=800$, not $14$'' clearer.

---

### Official Review · Reviewer_FkH1 · 2026-03-12

**Soundness:** 3
**Presentation:** 4
**Significance:** 3
**Originality:** 3
**Overall Recommendation:** 5
**Confidence:** 3

**Summary:**

This work proposes a new method called Priority-Aware Shapley Value (PASV), which extends Precedence Shapley Value (PSV) by Faigle and Kern (1992) as well as Weighted Shapley Value (WSV) by Kalai and Sammet (1987) and Nowak and Radzik (1995).
The authors also provide an axiomatization of this new method: Efficiency (E) + Linearity (L) + Null Player (NP) + Monotonicity (M) + Maximal Support (MS) => Precedence Random Order Value (PROV), and
Precedence Random Order Value (PROV) + State-Choice Factorization (SCF) + Weight Proportionality (WP) + Equal-Weight Uniformity (EWU) => Priority-Aware Shapley Value (PASV).
The authors propose an algorithm to evaluate the Priority-Aware Shapley Value through a Markov Chain Monte Carlo (MCMC) strategy based on an adjacent-swap Metropolis-Hastings sampler. Some experiments on data valuation and feature attribution are provided.

**Compliance With Llm Reviewing Policy:**

Affirmed.

**Key Questions For Authors:**

1. What's the complexity/scalability in practice of your method? In particular, in massive graphs or player sets?
2. How can users determine when they've drawn sufficient samples for reliable PASV estimates in their specific application?

**Limitations:**

yes

**Strengths And Weaknesses:**

This work has significant strengths in both theoretical depth and practical utility. The proposed method Priority-Aware Shapley Value (PASV), combines precedence constraints with soft priority weights within a coherent framework, addressing a gap in the Shapley value literature. The axiomatic characterization provides a strong theoretical grounding, while the MCMC-based computation offers a scalable solution to a non-trivial sampling problem. The priority sweeping diagnostic stands out as a particularly valuable contribution, transforming PASV from a point estimate into a sensitivity analysis tool for practitioners.
Overall, the paper appears to make solid contributions, though the practical impact will depend on how well these theoretical advances can be translated into deployable tools.

---

> ### Author Rebuttal · Authors · 2026-03-31
>
> Thank you for your encouraging review. Rebuttal figures:
>
> https://anonymous.4open.science/r/PASV-rebuttal/figures.pdf
>
> We added the following new experiments:
>
> - **RE1-1**, **RE1-2** in reply to aw3s;
> - **RE2** in reply to mTAt;
> - **RE4** in reply to uGr4;
> and their results will be reported as the narration goes. For detailed experiment set up, see our replies to Reviewers aw3s and uGr4.
>
> **Q1(a): Practical complexity:**
>
> We start with reviewing our computation procedure, which has **two separate parts**:
>
> 1. **Sampling linear extensions (MCMC burn-in / mixing).**
> - Each adjacent-swap M-H step uses only local information around the chosen swap position, so one iteration costs $O(d_{\rm max})\le O(n)$, where $d_{\max}$ is the maximum total degree in the DAG.
> - We added a new mixing experiment (Figure R1-1). Empirically, the mixing time is close to $O(n^2)$ across several poset families, well below the worst-case $O(n^3\log n)$ bound, and non-uniform $\lambda$ has limited effect.
>
> 2. **Estimating PASV once permutations are sampled.**
>
> - With $N_{\rm MC}$ retained permutations, the Monte Carlo stage needs $O(n{N_{\rm MC}})$ utility evaluations in total.
> - In practical implementation, repeated subsets can be cached and reused (avoid recomputing $U(S)$ for the same input $S$).
> - Total runtime is: (number of unique utility evaluations) $\times$ (cost per utility call) $+$ MCMC  overhead; Figure R1-2 reports these terms separately.
>
> **Q1(b): Scalability (number of players)**:
>
> - Our original paper already includes a data-valuation experiment with $\boldsymbol{n=800}$ **players** (Section 5.1);
> - In added experiment **RE1-2** (Figure R1-2), our original estimator scales to at least $\boldsymbol{n=8192}$;
> - We also devised a new regression-MSR proxy variant following the advice of Reviewer uGr4. This method can handle at least $\boldsymbol{n=800}$ and often improves accuracy under the same total budget.
>
> To calibrate scale, below is a compact comparison to representative prior estimators ([1]–[4] as in Reviewer uGr4’s comments):
>
> | method | largest reported \#players |
> |---|---:|
> | Regression-adjusted MC [1] | 101 |
> | KernelSHAP [2] | 100 |
> | Castro et al. permutation sampling [3] | 1000 |
> | Data Banzhaf / MSR [4] | 2000 |
> | **PASV (our method 1, in paper)** | **800** |
> | **PASV (our method 1, new experiment)** | **8192** |
> | **PASV (our method 2, new experiment)** | **2048** |
>
> So, while exact PASV is still combinatorial in nature, the empirical evidence suggests that the proposed estimator is **not limited to the small feature-attribution example** and remains practical for player sets in the **thousands**.
>
> **Q2: How can a user tell that enough samples have been drawn?**
>
> This is inherently **application-dependent**: some users care about accurate values, others only about ranking or top-$k$ decisions. To address this question empirically, we added Figure R1-2, which plots estimation error against the number of sampled permutations.
>
> Two takeaways:
> 1. In our benchmarks, the estimate usually becomes useful after **a few hundred** retained permutations; beyond that, improvement continues but with diminishing returns.
> 2. The most practical stopping rule is to monitor **stability of the quantity the user actually cares about**.
> A simple procedure we recommend is (related to Ghorbani & Zou, (2019)):
> - Run the estimator at budgets $m,2m,4m,\dots$;
> - Compare successive estimates via
> $$
> \Delta(m)=\frac{\\|\widehat\psi^{(2m)}-\widehat\psi^{(m)}\\|_2}{\\|\widehat\psi^{(2m)}\\|_2};
> $$
> - Also check whether the induced ranking / top-$k$ set has stabilized;
> - Optionally repeat with a few random seeds to confirm the same conclusion.
>
> If both the values and the downstream decision (e.g., ranking, top-$k$, sign pattern) stop changing materially, then additional samples are unlikely to change the conclusion. We will add this practical guidance to the revision.
>
> Thank you again for highlighting this point — we agree it is central for deployment, and your question helped us strengthen the paper substantially.
>
> References:
> A. Ghorbani and J. Zou (2019): ICML’19 pp. 2242-2251

---

> > ### Author Rebuttal · Reviewer_FkH1 · 2026-04-02
> >
> > Thank you.

---

> > > ### Author Response · Authors · 2026-04-02
> > >
> > > Thank you!

---

### Official Review · Reviewer_mTAt · 2026-03-13

**Soundness:** 3
**Presentation:** 3
**Significance:** 2
**Originality:** 2
**Overall Recommendation:** 4
**Confidence:** 4

**Summary:**

This paper introduces the Priority-Aware Shapley Value (PASV), a novel data valuation framework designed to integrate both hard precedence constraints (DAG) and soft priority weights ($\lambda$). The authors propose the State-Choice Factorization (SCF) and a localized adjacent-swap MCMC sampling algorithm. Furthermore, the authors explore the application of priority sweeping as a diagnostic tool for sensitivity analysis. Rigorous axiomatic proofs and comprehensive experiments in data valuation and feature attribution settings demonstrate the theoretical soundness and algorithmic efficiency.

**Compliance With Llm Reviewing Policy:**

Affirmed.

**Final Justification:**

The rebuttal addressed my main concerns, thus changing my evaluation.

**Key Questions For Authors:**

1. Regarding the weight $\lambda$ in the PASV: Could you provide a concrete, automated method to determine the weight vector $\lambda$ in a real-world scenario?
2. Regarding the parameterization $\lambda = b^c$ in the data valuation experiments: Could you explain the intuitive meaning of this specific function form in practical applications?
3. Regarding the MCMC algorithm: The paper claims efficient sampling via adjacent swaps but does not explicitly provide the time complexity analysis. What is the exact time complexity for this sampling algorithm?

**Limitations:**

The authors should explicitly consider the assumption of an oracle weight vector $\lambda$ as a major limitation. The lack of an endogenous or data-driven method to generate these weights limits the framework's deployment in trustless machine learning ecosystems.

**Strengths And Weaknesses:**

Strengths:
1. The motivation of this article is clear, and the challenge of extending the Shapley value to accommodate both hard precedence constraints (DAG) and soft subjective preferences simultaneously is a non-trivial problem in cooperative game theory and data valuation.
2. The theoretical foundation proposed in this article is rigorous. The axiomatic introduction of SCF provides a solid theoretical justification for the PASV framework.
3. The article utilizes an adjacent-swap Markov Chain Monte Carlo sampler, which is cleverly designed.

Weakness:
1. This method relies heavily on prior information about the weight vector $\lambda$. Without an automated or data-driven method to determine $\lambda$, PASV remains merely a closed-loop mathematical exercise rather than a deployable tool.
2. The paper does not provide a robustness analysis regarding DAG misspecification. It remains unclear how sensitive the PASV scores are to a single missing or mistaken edge, which is a critical reliability concern for deployment.

---

> ### Author Rebuttal · Authors · 2026-03-31
>
> Thank you for your comments. Rebuttal figures:
>
> https://anonymous.4open.science/r/PASV-rebuttal/figures.pdf
>
> **Q1: you commented: "This method relies heavily on prior information about the weight vector . Without an automated or data-driven method to determine $\lambda$, PASV remains merely a closed-loop mathematical exercise rather than a deployable tool."**
>
> Allow us to clarify that:
> 1. **Our method is fully operational with or without $\lambda$.**
>
> 	a. **Our priority sweeping tool exactly addresses the realistic setting where the metadata needed for determining $\lambda$ is missing.**
>
> 	We can start with $\lambda_i\equiv 1$, reducing to PSV. Then priority sweeping tells us whether the valuation/ranking is robust to the missing $\\lambda$ information (or whether the ignored $\lambda$ materially matters).
>
> 	b. **It is not difficult to practically compute $\\lambda$, when the corresponding information is available.**
>
> 	For instance, $\lambda_i$ can be a function of compute cost, compliance risk, or originality score. But these key metadata are usually missing in real-world datasets – **this limitation is not in our methodology, but in the current data collection practice**.
>
> 2. **Methods that “do not require $\lambda$” ignored the issue rather than solved it.**
>
> 	The $\lambda$ vector can encode important information on trustworthiness, originality, risk, etc. Previous methods such as PSV simply ignored $\lambda$ by setting $\lambda_i\equiv 1$; this does not remove the issue. In contrast, **our work makes this hidden assumption explicit** and **calls for attention to this important aspect of the problem**.
>
> Therefore, incorporating $\lambda$ is **not a weakness** of our method, but a **highlighted contribution**.
>
> **Q2: sensitivity analysis for DAG misspecification**
>
> Thank you for this important suggestion!
>
> We add:
>
> **Added experiment RE2**
>
> based on our feature-attribution experiment.
>
> Setup:
>
> - DAG: general DAG in Figure 6(b) (12 nodes, 26 edges)
> - Weights: $\lambda\equiv 1$
> - Perturb edges only, not nodes; all other configurations unchanged
>
> Notation:
>
> - $\psi, \psi’$: PASV vectors under original/perturbed DAG
> - $Q_{ij}, Q’_{ij}$: pairwise ordering probabilities estimated via Monte Carlo
>
> Metrics (2–5 each plotted against RAS in Figure R2):
>
> 1. **Relative Attribution Shift (RAS)**: $\\|\psi’-\psi\\|_2/\\|\psi\\|_2$
> 2. **Pearson correlation** between $\psi$ and $\psi’$
> 3. **Top-4 overlap**: number of shared nodes among the 4 highest-valued under $\psi$ vs. $\psi’$
> 4. **Order distribution change**: mean $\|Q’\_{ij}-Q\_{ij}\|$ over all pairs
> 5. **Relation change**: fraction of node pairs whose deterministic precedence relation ($i\prec j$, $j\prec i$, or incomparable) differs between the two DAGs
>
>
> Scenarios:
>
> 1. Single-edge deletion. Only consider 12 effective deletions (transitive reduction).
> 2. Single-edge addition. Randomly sampled 20 valid additions among currently incomparable, acyclic pairs.
> 3. DAG disconnection. Minimum-edge-cut deletions splitting all nodes into two equal components; 6 cases identified.
> 4. Delete/Keep top-K influential edges. Rank the 12 effective edge deletions by their RAS. For $K\in\\{3,4,5\\}$, plotted: (a) delete the top-$K$, or (b) keep only top-$K$ & remove all others.
>
> Results in Figure R2:
>
> We highlight a key point: **robustness to DAG change is not always a desirable property.**
>
> - Consider: $i$ and $j$ both claim a major discovery; then a good method **should produce very different results** for the three cases: $i\to j$, $j\to i$ or no edge.
> - Therefore, the right criterion is whether the sensitivity of the evaluation method **aligns with the semantic consequence of the DAG change**.
>
> Our results confirm exactly this: PASV is **robust to inconsequential perturbations** and **appropriately sensitive to semantically significant ones**.
> - Minor structural changes (adding/deleting a single non-critical edge) leave PASV nearly unchanged (RAS<0.05, Pearson>0.97), while drastic changes (disconnection, removing critical edges, removing most edges) produce larger shifts.
> - The underlying driver: across all scenarios, RAS is tightly correlated with how much the perturbation alters the node ordering distribution.
>
> **Q3: intuition behind $\lambda = b^c$**
> - $c$ assigns each player a priority *tier*
> - $b$ controls the separation strength between tiers.
> E.g., for $b>1$, setting $c=(0,1,0,2,2)$ puts Owner/Boosters at baseline ($\lambda=1$), mildly deprioritizes Anchor ($\lambda=b$), strongly deprioritizes Copier/Poisoner ($\lambda=b^2$).
>
> Further examples:
> - $b=1$ recovers PSV;
> - $b\to\infty$ approaches hard precedence (Prop. 3.13–3.14).
>
> **Q4: MCMC time complexity**
>
> See our reply to Reviewer aw3s for full discussion.
>
> Gist:
>
> - Each M-H iteration costs $O(d_{\max})$, where $d_{\max}$ is max DAG degree (in-edges + out-edges); this is no more than $O(n)$.
> - Empirical mixing time: roughly $O(n^2)$, well below the worst-case $O(n^3\log n)$.
> - Non-uniform weights do not worsen mixing.

---

> > ### Author Rebuttal · Reviewer_mTAt · 2026-04-03
> >
> > Thanks for your responses. I'd like to raise the score to 4.

---

> > > ### Author Response · Authors · 2026-04-03
> > >
> > > Thank you!

---

### Official Review · Reviewer_aw3s · 2026-03-13

**Soundness:** 3
**Presentation:** 3
**Significance:** 3
**Originality:** 3
**Overall Recommendation:** 4
**Confidence:** 2

**Summary:**

The paper addresses limitations in the standard shapley valuation in particular since some points have precedence over others such as in data augmentation and causal features. Specifically, the paper proposes priority aware Shapley value where essentially points are weighted according to their priority; based on axioms that incorporate weights and priority. The paper proposes an algorithm for computing the priority based Shapley values. Experiments are done on a collection of datasets.

**Compliance With Llm Reviewing Policy:**

Affirmed.

**Key Questions For Authors:**

Same as point under weaknesses. Particularly, the runtime and the achieved approximation?

**Limitations:**

Yes, I think so.

**Strengths And Weaknesses:**

Strengths:
-I think the motivation behind the paper is very sound
-the paper gave a clean and nice solution to the problem based on axioms

Weaknesses:
-The main thing not clear to me in the paper is that it does not seem to give a full discussion on the computation of PASV and its approximation. Specifically, what is the run-time and using the sampling approach how good is the approximation?

---

> ### Author Rebuttal · Authors · 2026-03-31
>
> We appreciate your comments.  Rebuttal figures:
>
> https://anonymous.4open.science/r/PASV-rebuttal/figures.pdf
>
> **Q1: runtime and approximation accuracy?**
>
> **Big picture of our method’s computation:** two stages:
> 1. Burn-in: run MCMC until stationarity; no utility evaluation; chain outputs discarded.
> 2. Take MCMC samples with thinning; for each sampled permutation, evaluate utility function; properly average them to estimate PASV.
>
> Your question thus contains two distinct aspects (addressed separately):
> - Stage 1 cost depends on the **mixing time** of MCMC (how many iterations the chain requires to reach stationarity).
> - For stage 2, the key quantity is $N_{\rm MC}$, the **number of sampled permutations** for Monte Carlo. Total utility evaluations: $O(nN_{\rm MC})$.
>
> There are two versions of our method:
> - [Our method 1] The algorithm in our original submission.
> - [Our method 2] A new “quadratic regression MSR” proxy method inspired by Reviewer uGr4.
>
> They’re the same for stage 1, only differ in stage 2;
>
> **Q1(a): MCMC mixing time**
>
> We add:
>
> **Added experiment RE1-1.**
>
> Define:
> - $P_{\pi_0}^t$: chain’s law after $t$ iterations starting from permutation $\pi_0$;
> - $P^\*$: stationary distribution;
> - Mixing quality:
> $$
> D_t=\max_{i,j\in[n]}\left|P_{\pi_0}^t(i \prec j\ \ \text{in }\pi)-P^*(i \prec j\ \ \text{in }\pi)\right|. \tag{R1-1}
> $$
>
> Set up:
>  - $n\\in\\{5,10, …,25\\}$;
>  - Four poset scenarios from Talvitie et al (2017); two sub-settings each (see Figure R1-1).
>  - iid $\lambda_i\sim{\rm Unif}[1,R]$; $R$ controls non-uniformity level (see Figure R1-1).
>  - We run 10000 independent chains to estimate $P_{\pi_0}^t$ in Eq. (R1-1), and obtain the true $P^*$ via dynamic programming (Algorithm R1 of Kangas et al (2016), adapted for our non-uniform setting).
>
> **Empirical mixing time** = the earliest $t$ s.t. $D_t<1/4$.
>
> Result in Figure R1-1:
>
> Reference lines:
> - Theoretical worst-case mixing time: $O(n^3\log n)$
> - Empirically, $O(n^2)$ from Talvitie et al (2017)
>
> Results:
> - Our MCMC mixes reasonably fast, roughly aligning with $O(n^2)$ mixing time in Talvitie et al (2017);
> - Non-uniformity of $\lambda_i$’s has limited impact;
> - Mixing time is often not the main computational bottleneck (utility evaluation is).
>
> **Q1(b): how many permutations do we need to sample to well-approximate PASV?**
>
> We add:
>
> **Added experiment RE1-2**
>
> with sum-of-unanimity (SOU) utility:
> $$
> U(S)=\sum_{j=1}^d \alpha_j \mathbf{1}\\{T_j\subseteq S\\}, \quad T_j\subseteq [n],
> $$
> where iid $\alpha_j\sim{\rm Unif}[0.5,1.5]$; $(d,T_j)$ specified below.
>
>
> Scenario 1:
> - DAG: $\\{B_k\\}\_{k=1}^K$ uniformly partitions $[n]$. Treating $B_k$’s as nodes, add edge $B_k\to B_\ell$ ($k<\ell$) w.p. 0.8; among $n$ players, $i\to j$ iff $i\in B_k$, $j\in B_\ell$ with $B_k\to B_\ell$, and incomparable otherwise.
> - $\lambda\equiv 1$; $d=K$; $T_j=B_j$ for $j\in[d]$.
> - True PASV: $\psi_i(U)=\alpha_j/|T_j|$ if $i\in T_j$.
>
> Scenario 2:
> - DAG: Ordered partition $B_1 \\prec \\dots\\prec B_K$ where $\\{B_k\\}_{k=1}^K$ uniformly partitions $[n]$.
> - Three cases: $\lambda\equiv 1$, $\lambda\sim\mathrm{Unif}[1,10]$, $\lambda\sim\mathrm{Unif}[1,100]$.
> - $d=n^2$; $T_j$ sampled uniformly from $2^{[n]}$.
> - True PASV: $\psi_i(U)=\sum_{j:i\in M_{T_j}}\alpha_j\frac{\lambda_i}{\sum_{r\in M_{T_j}}\lambda_r}$ where $M_T=T\cap B_{r(T)}$, $r(T)=\max\\{k:T\cap B_k\neq\emptyset\\}$.
>
> Shared settings for both scenarios:
> - $n\in2^{\\{7,9,11,13\\}}$;
> - $K=n/16$;
> - $N_{\rm MC}=10^4$;
> - burn-in: $10^6$; thinning: $10^3$.
>
> Our method 1 operates as in our paper. We cache $U(S)$ for each newly encountered subset $S$ and reuse upon revisit.
>
> Metrics:
> - Absolute relative error (ARE):
>
> $$
> \\mathrm{ARE}(m)=\\frac{\\|\\widehat{\\psi}\^{(m)}-\\psi\^\\star\\|_2}{\\|\\psi^\\star\\|_2},
> $$
>
> where define $\psi^\star \in \mathbb{R}^n$: true PASV value; $\widehat{\psi}^{(m)}$: estimate with $m$ sampled permutations.
>
> - Area under convergence curve (AUCC):
> $$
> \frac{1}{100}\sum_{\ell=1}^{100}\mathrm{ARE}(100\ell)
> $$
> - Number of unique utility evaluations
> - Runtime: separately for utility evaluation and everything else.
>
> Result for our method 1 in Figure R1-2. Key takeaways:
> 1. Across all settings, our method 1 well-approximates the true PASV for $N_{\rm MC}$ at a few hundreds.
> 2. Our method comfortably scales up to massive graphs with a few thousands of players at moderate runtime cost. This is comparable to the scalability in the representative works as references [1]-[4] cited by Reviewer uGr4. For a detailed comparison table, see our reply to uGr4.
> 3. Caching $U(S)$ reduces total number of utility calls.
> 4. In real-world applications where utility evaluation is costly, users can predict runtime by: (# utility calls at their scale) $\times$ (cost per evaluation) + (MCMC runtime from our figure).
>
> Result for our method 2: see our reply to Reviewer uGr4.
>
> **References:**
> - K. Kangas et al (2016): IJCAI’16 pp. 603-609
> - T. Talvitie et al (2017): IJCAI’17 pp. 524-530

---

> > ### Author Rebuttal · Reviewer_aw3s · 2026-04-06
> >
> > While the authors have provided a characterization of the run-time which I think is a very good addition for the paper. Run-times of order $n^3$ and $n^2$ are still very large. I will maintain my score.

---

> > > ### Author Response · Authors · 2026-04-06
> > >
> > > Thank you for the follow-up. However, the comment **conflates the stage 1 mixing iterations with the end-to-end runtime of PASV estimation**. We clarify below.
> > >
> > > 1. **What the runtime actually is**
> > >
> > >    * Stage 1 (MCMC burn-in): each iteration is a **local adjacent-swap proposal of two indices, using only local information; no utility call; cheap**
> > >    * Stage 2 (Monte Carlo): each iteration requires **one utility call; expensive**
> > >    * End-to-end runtime:
> > >    $$
> > >    \text{Runtime} = \underbrace{{(\text{MCMC overhead})}}\_{{\text{Stage 1}}} + \underbrace{{(\text{\\# utility calls}) \times (\text{cost per utility call})}}\_{{\text{Stage 2}}}
> > >    $$
> > >
> > > 2. **Stage 1 is not the bottleneck.**
> > >
> > >    * The key scalability lever is the **number of utility calls in stage 2**, not the stage 1 mixing iterations in isolation. Citing $O(n^2)$/$O(n^3)$ alone does not capture the full picture.
> > >    * The reported $O(n^2)$ mixing time is **optimal** for this type of MCMC, according to [Bubley & Dyer, 1999] and [Talvitie et al, 2017].
> > >       * Intuition: define a “round” as enough swaps for most players to have had the opportunity to update their positions – this requires $O(n)$ swaps (since there are $n$ positions).  The chain needs $O(n)$ such rounds to mix well.
> > >       * The $O(n^2)$ total swaps is a **minimal requirement from MCMC theory** rather than an inefficiency of our method.
> > >
> > > 3. **Stage 2 controls scalability, and our method handles it competitively.**
> > >
> > >    * The key lever is the **number of utility calls** $N_{\rm MC}$.
> > >    * In our added experiments, accurate PASV estimates already emerge with $N_{\rm MC}$ on the order of a few hundred (Figure R1-2).
> > >    * Our stage 2 (and overall) scalability is **head-to-head with the representative existing scalable estimators**. The comparison table in our reply to Reviewer FkH1 shows scaling to $n=8192$ players, on par with or exceeding the largest player counts reported in the representative prior works [1]--[4] cited by Reviewer uGr4.
> > >
> > > 4. **MCMC is inherent to the problem**
> > >    * PASV is axiomatically defined through a permutation distribution $p^{(\\preceq,\\lambda)}$ given by an unnormalized probability mass function. All users who adopt PASV's axioms and SCF would uniquely target this distribution (our Theorem 3.12).
> > >    * Adjacent-swap MCMC is the standard practical method for sampling permutations from this $p^{(\\preceq,\\lambda)}$. Even PASV's special case PSV relies on the same MCMC machinery for general DAGs.
> > >
> > > **In summary**, our rebuttal explicitly separates stage 1 (cheap; no utility calls) from stage 2 (expensive; utility calls) with detailed added experiments. Therefore, **citing the stage 1 iteration count alone does not accurately characterize our method’s runtime**. The end-to-end scalability of our method is demonstrated empirically and is competitive with representative scalable estimators.
> > >
> > > **References:**
> > >
> > > [Bubley & Dyer, 1999] R. Bubley & M. Dyer. Faster random generation of linear extensions. _Discrete Mathematics_, 201(1):81–88, 1999
> > >
> > > [Talvitie et al, 2017] T. Talvitie, T. Niinimaki and Mikko Koivisto. The Mixing of Markov Chains on Linear Extensions in Practice. _IJCAI_, pp. 524-530. 2017.

---

### Decision · Program_Chairs · 2026-04-30

**Decision:**

Accept (regular)

**Comment:**

After the discussion the reviews range from mildly to solidly positive.  The reviewers and I particularly appreciate the theoretical formulation of the approach and agree that there is also enough evidence provided on the empirical / practical side. Given this the paper seems a solid candidate for inclusion in the program.